# Vulnerability of short-term memory in a mouse model of Alzheimer's disease

Chunyue Li[1,3], Xin Wei Chia [1,3], Guozhong Xu[1,3], Lee Fang Ang[1] & Hiroshi Makino [1,2] ✉

Interference from distracting stimuli renders short-term memory vulnerable. While behavioral evidence suggests short-term memory deficits in Alzheimer's disease (AD), the underlying neural mechanisms remain poorly understood. Using a mouse model of AD (APP-KI), we identified increased susceptibility of short-term memory to sensory perturbations. Simultaneous two-photon calcium imaging across eight cortical regions during a delayed-response task showed that distractors disrupted neural selectivity at both single-neuron and population levels in APP-KI mice. Recurrent neural network models replicating the neural activity of APP-KI mice exhibited decreased stability, consistent with reduced functional connectivity across the dorsal cortex. Furthermore, analyses of multi-regional corticocortical communication revealed reduced spatiotemporal degeneracy in activity transmission within the dorsal cortex of APP-KI mice, which could account for their attenuated robustness during sensorimotor transformations. Collectively, these findings identify reduced functional connectivity and impaired spatiotemporal degeneracy as central mechanisms of short-term memory deficits in the APP-KI mouse model of AD.

Alzheimer's disease (AD) is an age-related neurodegenerative disorder characterized by progressive cognitive decline[1]. Early symptoms include short-term memory deficits, followed by gradual impairments in both declarative and non-declarative memory[2]. The primary neuropathological hallmarks are extracellular amyloid-β (Aβ) plaques and intracellular neurofibrillary tangles composed of hyperphosphorylated tau[3–5]. Numerous animal models have been developed to elucidate the etiology and progression of the disease[6,7]. Among these, amyloid precursor protein knock-in (APP-KI) mice, which carry the mutated human $App^{NL\text{-}G\text{-}F}$ gene (NL: Swedish; F: Beyreuther/Iberian; G: Arctic mutations) inserted at the endogenous mouse $App$ locus, exhibit age-dependent acceleration of Aβ deposition, increased gliosis, synaptic dysfunction and cognitive impairments[8–10]. These models have been extensively characterized, with a growing body of research identifying neural activity mechanisms that link molecular alterations to corresponding behavioral phenotypes[11–15].

Short-term memory, defined as the ability to maintain information online, is essential for motor planning. During memory tasks, neurons exhibit sustained activity in response to brief stimuli[16–21]. This persistent activity arises from recurrent positive feedback loops within both local and distributed brain networks[22–27]. Short-term memory deficits are evident in patients with AD and replicated in mouse models[28,29]. However, the precise disruptions in recurrent functional connectivity that support persistent neural activity in AD remain unclear.

Most brain functions depend on the flexible routing of neural information across distributed networks[30,31]. Although anatomical connectivity provides the structural scaffold, it does not fully determine communication schemes. The concept of a "communication subspace" posits that only specific neural activity patterns within a source region effectively propagate to a target region, forming a selective communication channel, while non-aligned patterns fail to transmit[32,33]. This framework has been explored in sensory processing and sleep[33,34]. Whether it is disrupted in animal models of AD, however, remains unknown.

Degeneracy is a fundamental principle observed across a wide range of biological systems[35]. In neuroscience, it refers to the existence of multiple parallel neural pathways capable of generating the same

[1]Lee Kong Chian School of Medicine, Nanyang Technological University, Singapore, Singapore. [2]Department of Physiology, Keio University School of Medicine, Tokyo, Japan. [3]These authors contributed equally: Chunyue Li, Xin Wei Chia, Guozhong Xu. ✉e-mail: hmakino@keio.jp

output, function or behavior. By providing alternative routes for information processing, degeneracy confers robustness and flexibility, ensuring that neural functions can persist even when one pathway is compromised, sometimes at the expense of specificity and metabolic efficiency[36]. In AD, reduced degeneracy in neural communication may underlie short-term memory deficits, leading to attenuated robustness against perturbations[37].

To investigate how short-term memory deficits arise in AD and their neural substrates, we simultaneously recorded neural activity across multiple cortical regions in APP-KI mice that are critical for different phases of sensorimotor transformations. Our empirical and theoretical analyses show that APP-KI mice exhibit heightened vulnerability in short-term memory, associated with unstable cortical activity dynamics. We propose that this instability stems from reduced functional connectivity and diminished spatiotemporal degeneracy in sensorimotor information processing, ultimately undermining the robustness of short-term memory in AD.

## Results

### Two-photon calcium imaging of mouse dorsal cortex during a delayed-response task

We trained head-restrained APP-KI (APP-KI × Thy1-GCaMP6f, $n = 11$) and control (Thy1-GCaMP6f, $n = 7$) mice, both expressing the calcium

indicator GCaMP6f[38], in a delayed-response task to assess short-term memory and sensorimotor processing. In this task, mice received a brief tactile stimulus on either the left or right whiskers (brass rods sweeping at ~20 Hz for 1 s), with trial types randomly interleaved. After a 4-s delay, both the left and right water spouts were presented and mice were cued (4 kHz, 0.2 s) to respond by licking in the direction corresponding to the stimulated side (Fig. 1a). During the delay, mice relied on short-term memory to maintain the sensory information and guide their upcoming action. Correct responses were rewarded with 4 µl of sucrose solution (10–15% concentration), while incorrect responses were punished with 0.5 s of white noise and a 6–8 s timeout. Under these conditions, baseline task performance did not significantly differ between control and APP-KI mice ($P = 0.20$, $n = 32$ and 40 sessions for control and APP-KI mice, respectively, one-tailed bootstrap, Figs. 1b, S1a, b). These results suggest that, under baseline conditions, sensorimotor processing and short-term memory functions in APP-KI mice remained largely intact.

We examined task-related neural responses in control and APP-KI mice (6–10 months old) by simultaneously recording the activity of individual layer 2/3 pyramidal neurons across eight cortical regions in the left hemisphere. This age range was selected to ensure that all APP-KI mice had reached a stage of robust and stable Aβ plaque burden[8] (Figs. 1c, S1c and Table S1), during which no significant behavioral

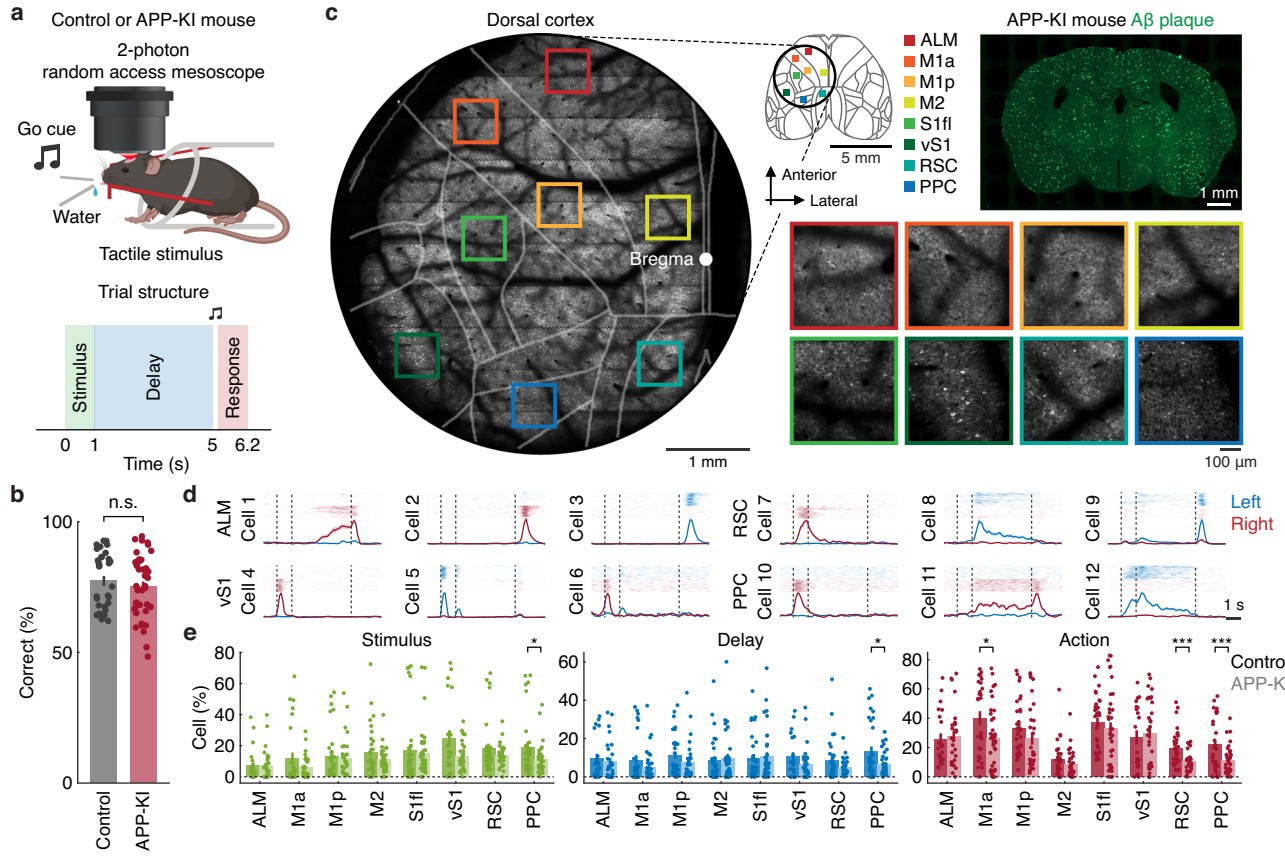

**Fig. 1 | Two-photon calcium imaging of the mouse dorsal cortex during a delayed-response task. a** Schematic of the delayed-response task (top, created in BioRender. Makino, H. (2026) https://BioRender.com/a5h84gh) and its trial structure (bottom). **b** Task performance in control and APP-KI mice (n.s., $P = 0.20$, $n = 32$ and 40 sessions for control and APP-KI mice, respectively, one-tailed bootstrap). Error bars indicate mean ± SEM. **c** Example of two-photon calcium imaging of the dorsal cortex and Aβ plaques in an APP-KI mouse. Individual neurons across eight cortical regions were simultaneously imaged during the task. ALM: anterior lateral motor cortex; M1a, M1p: anterior and posterior regions of primary motor cortex; M2: secondary motor cortex; S1fl, vS1: primary somatosensory cortex for forelimb and vibrissae; RSC: retrosplenial cortex; PPC: posterior parietal cortex. Two-photon

calcium imaging data and Aβ plaque images were acquired across all mice and from 6 mice, respectively. **d** Examples of task-related neural activity during correct trials for two trial types across cortical regions in an APP-KI mouse. Top. Trial-by-trial activity. Bottom. Mean neural activity. Vertical dashed lines separate the stimulus, delay and action epochs. Shaded areas represent mean ± SEM. **e** Proportion of neurons exhibiting task epoch-specific selectivity, computed from correct trials (*$P < 0.05$, ***$P < 0.001$, control: $n = 22, 27, 32, 30, 32, 21, 32, 31$ sessions; APP-KI: $n = 29, 37, 30, 33, 36, 31, 31, 38$ sessions for ALM, M1a, M1p, M2, S1fl, vS1, RSC and PPC, respectively, one-tailed bootstrap with an FDR using the Benjamini-Hochberg procedure). Error bars indicate mean ± SEM. Source data are provided as a Source Data file.

differences were observed (Fig. S1d). The recorded regions included the anterior lateral motor cortex (ALM), anterior and posterior subdivisions of primary motor cortex related to tongue and forelimb movement (M1a and M1p), secondary motor cortex (M2), primary somatosensory cortex for the forelimb and vibrissae (S1fl and vS1), retrosplenial cortex (RSC) and posterior parietal cortex (PPC). Neural activity was recorded using a two-photon random access mesoscope[39] (Fig. 1c). Consistent with previous reports of distinct trial-type selectivity patterns across cortical regions during delayed-response tasks[40,41], we observed heterogeneous neural response profiles throughout these regions in both APP-KI and control mice (Fig. 1d). Notably, APP-KI mice showed a reduced proportion of neurons with stimulus, delay or action selectivity, with the most pronounced differences in the PPC (stimulus: $P < 0.05$; delay: $P < 0.05$; action: $P < 0.001$, $n = 31$ and 38 sessions for control and APP-KI mice, respectively, one-tailed bootstrap with a false discovery rate (FDR) using the Benjamini-Hochberg procedure, Fig. 1e and S1e). This reduction in the PPC was not attributable to differences in the number of Thy1–GCaMP–positive cells in the control and APP-KI mice (stimulus: control: $R^2 = 0.13$, $P > 0.05$; APP-KI: $R^2 = 0.07$, $P > 0.05$; delay: control: $R^2 = 0.08$, $P > 0.05$; APP-KI: $R^2 = 0.18$, $P > 0.05$; action: control: $R^2 = 0.14$, $P > 0.05$; APP-KI: $R^2 = 0.12$, $P > 0.05$, $n = 31$ and 38 sessions for control and APP-KI mice, respectively, Pearson correlation with an FDR using the Benjamini-Hochberg procedure). In addition, we found a significant reduction in vS1 selectivity during the early delay period ($P < 0.05$, $n = 21$ and 31 sessions for control and APP-KI mice, respectively, one-tailed bootstrap with an FDR using the Benjamini-Hochberg procedure, Fig. S1e) together with a strong trend toward reduced vS1 selectivity during the stimulus period ($P < 0.01$, $n = 21$ and 31 sessions for control and APP-KI mice, respectively, one-tailed bootstrap without a multiple-comparison correction method, Fig. 1e). These results suggest that selectivity was already degraded at the input stage and further amplified in the PPC of APP-KI mice.

## Behavioral and neural vulnerability of short-term memory to sensory perturbations in APP-KI mice

Given the established role of the PPC in short-term memory[40,42–44], the reduced proportion of trial-type-selective neurons in the PPC of APP-KI mice suggests that their short-term memory deficits may become more apparent under more demanding conditions. To explore this, we introduced brief distracting stimuli (~20 Hz for 0.2 s) to both sides of the snout in ~30% of randomly interleaved trials, delivered at one of three time points during the 4-s delay (1 s, 2 s or 3 s after delay onset, each comprising ~10% of trials) (Fig. 2a). Compared with control mice, APP-KI mice were more behaviorally sensitive to these perturbations ($P < 0.05$, $P < 0.05$, $P < 0.001$ for early, middle and late distractors, respectively, $n = 32$ and 40 sessions for control and APP-KI mice, respectively, one-tailed bootstrap with Bonferroni correction, Fig. 2b). We detected no significant behavioral differences between younger (6-month-old) and older (9-month-old) APP-KI mice, suggesting comparable vulnerability to perturbations across the examined age range (Fig. S1d). These results indicate that short-term memory in APP-KI mice is more vulnerable to distraction.

The reduced robustness of short-term memory in APP-KI mice may be attributed to an impaired ability to maintain persistent neural activity throughout the delay period. To investigate the neural basis of this susceptibility, we focused on neurons that exhibited trial-type selectivity during the delay. In non-distractor trials, delay-selective activity was comparable between control and APP-KI mice (Fig. 2c). We identified the preferred trial type (left or right) during the delay for each neuron and subtracted task-related activity in left non-distractor trials from that in right non-distractor trials. This yielded upward and downward deflections for right- and left-preferring delay neurons, respectively. Trial-type selectivity was then quantified as the difference between the two neuron types. Both control and APP-KI mice showed

significantly stronger selectivity in correct compared with incorrect trials (Fig. 2d), suggesting that trial-type selectivity reliably serves as a neural substrate of correct performance.

When distractors were introduced, their impact on delay-selective neurons varied across cortical regions. In control mice, selectivity in anterior regions such as the ALM remained relatively stable, but this robustness was not evident in other areas (Fig. 2e, f, S2a, b). In contrast, APP-KI mice showed a more pronounced reduction in trial-type selectivity across most regions, with particularly strong effects in the ALM, M2 and PPC (Fig. 2e, f, S2a, b). Similar to delay-selective neurons, stimulus-selective neurons also exhibited reduced trial-type selectivity in APP-KI mice compared with controls, with the reduction being more pronounced in posterior regions such as vS1 and PPC, whereas the decrease in action-selective neurons was more moderate (Fig. S2b). These findings indicate that stimulus- and delay-selective neurons may overlap, whereas action-selective neurons constitute a distinct population. Together, these results identify neural substrates underlying the heightened vulnerability of short-term memory to distractors in APP-KI mice.

## Vulnerability of neural population dynamics in APP-KI mice during short-term memory

To assess neural vulnerability at the population level, we applied CEBRA (Consistent EmBeddings of high-dimensional Recordings using Auxiliary variables), a nonlinear dimensionality reduction method[45]. CEBRA extracts low-dimensional embedding spaces shared across animals and sessions, enabling behavioral and neural data to be represented in terms of task-relevant variables.

We first evaluated how distractors affected cortical population dynamics. To this end, we constructed "pseudo-mice" by sampling random subsets of neurons from eight cortical regions. CEBRA models were trained and evaluated on non-distractor trials using stimulus type as the behavioral label, confirming that both control and APP-KI mice encoded left- and right-trial information in distinct low-dimensional latent spaces (Fig. 3a and S3a). To rule out potential label supervision artifacts, we also trained fully unsupervised CEBRA models and observed a similar separation (Fig. S3b). When distractors were introduced, however, a k-nearest-neighbor (kNN) decoder applied to the CEBRA embeddings showed reduced accuracy in stimulus identification (Fig. 3b, c). Notably, APP-KI mice exhibited significantly higher sensitivity to these perturbations ($P < 0.001$ for early, middle and late distractors, $n = 25$ pseudo-mice for control and APP-KI mice, one-tailed bootstrap with an FDR using the Benjamini-Hochberg procedure, Fig. 3c). Regional analyses further revealed that ALM and M2 contributed most consistently to this heightened sensitivity in APP-KI mice across distractor conditions ($n = 25$ pseudo-mice for control and APP-KI mice, one-tailed bootstrap with an FDR using Benjamini-Hochberg procedure, Fig. 3d–f). These results were consistent across varying numbers of pseudo-mice (Fig. S3c–f).

Next, we trained CEBRA models with choice as the behavioral label to examine how distractors modulate choice-related neural dynamics. Similar to the stimulus-related embeddings, the choice-related low-dimensional trajectories distinguished left from right choices (Fig. 3g and S3g). Under distractor conditions, the kNN decoder again revealed increased vulnerability in APP-KI mice ($P < 0.001$ for early, middle and late distractors, $n = 25$ pseudo-mice for control and APP-KI mice, one-tailed bootstrap with an FDR using the Benjamini-Hochberg procedure, Fig. 3h, i). As with stimulus selectivity, ALM and M2 consistently exhibited pronounced sensitivity to perturbations in APP-KI mice across distractor conditions ($n = 25$ pseudo-mice for control and APP-KI mice, one-tailed bootstrap with an FDR using the Benjamini-Hochberg procedure, Fig. 3d–f). In addition, these results were robust across different numbers of pseudo-mice (Fig. S3h–k). Together, these findings demonstrate that both stimulus- and choice-related representations, as encoded by cortical

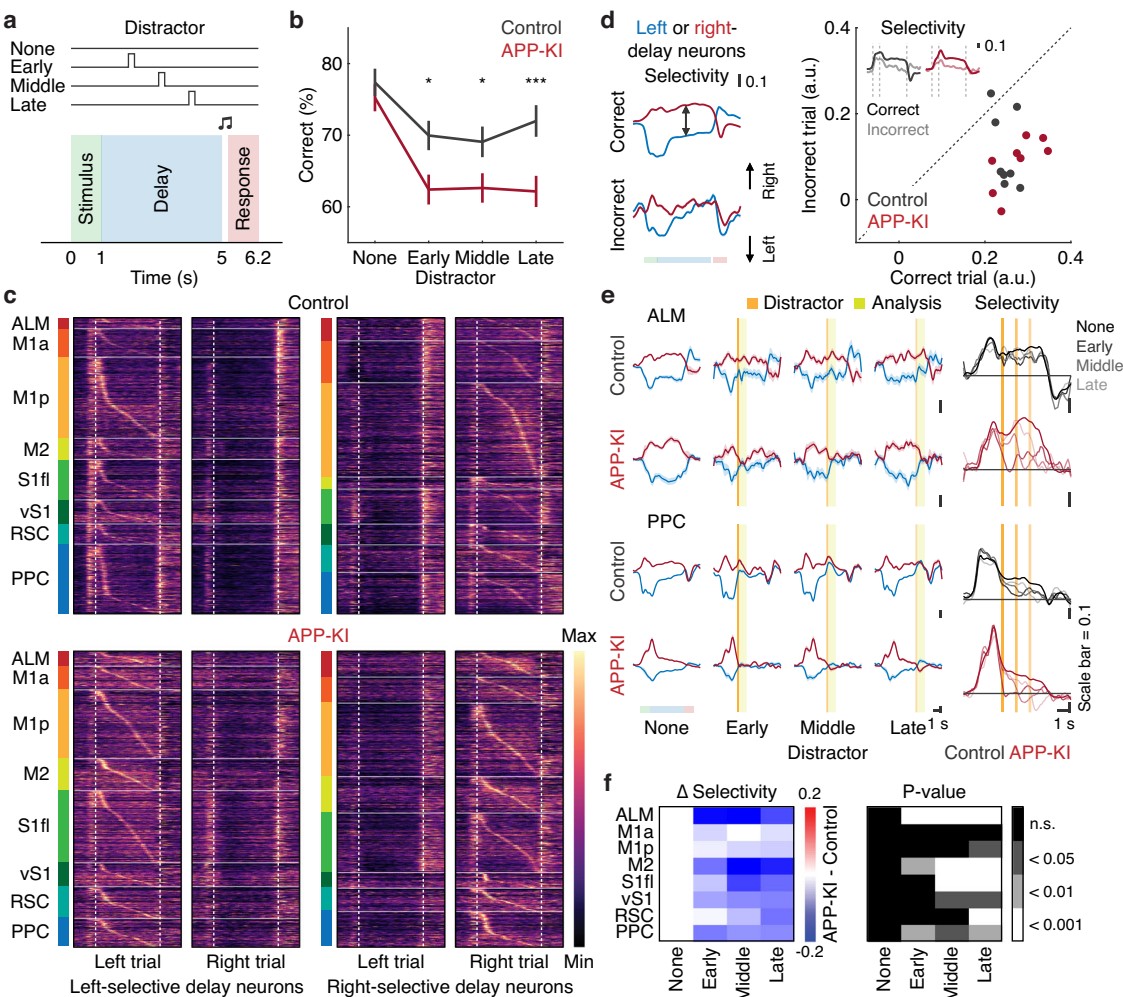

**Fig. 2 | Distractors-mediated reduction of trial-type selectivity in single neurons. a** Schematic of the trial structure with distractors. Distractors were presented at one of three time points (early, middle, late) in ~30% of trials (~10% each), randomly interleaved. **b** Task performance with distractors in control and APP-KI mice (***$P < 0.001$, *$P < 0.05$, $n = 32$ and 40 sessions for control and APP-KI mice, respectively, one-tailed bootstrap with Bonferroni correction). Error bars indicate mean ± SEM. **c** Normalized trial-averaged activity of left- and right-selective delay neurons during left and right correct non-distractor trials in control and APP-KI mice. Vertical dashed lines separate the stimulus, delay and action epochs. **d** Left. Mean task-related activity difference of delay neurons over time during correct and incorrect non-distractor trials in control mice. Task-related activity differences were computed by subtracting activity during left non-distractor trials from that during right non-distractor trials, resulting in upward and downward deflections for right- and left-preferring delay neurons, respectively. Trial-type selectivity is represented by the distance between the two activity traces. Right. Mean trial-type selectivity across the eight cortical regions during correct and incorrect non-

distractor trials in control and APP-KI mice. Inset indicates mean trial-type selectivity profiles of delay neurons over time in correct and incorrect non-distractor trials. Shaded areas represent mean ± SEM. **e** Left. Mean task-related activity differences of delay neurons in ALM and PPC, with and without distractors, in control and APP-KI mice, computed as in (**d**). Blue and red traces indicate activity of left- and right-delay neurons, respectively. Orange vertical lines indicate the period when the distractor was present, and green shaded areas mark the 1-s analysis window following each distractor. Right. Mean trial-type selectivity profiles of delay neurons over time for trials with and without distractors in control and APP-KI mice. In ALM, distractors disrupted trial-type selectivity specifically in APP-KI mice but not in controls. Shaded areas represent mean ± SEM. **f** Left. Differences in region-specific distractor-mediated modulation of trial-type selectivity between control and APP-KI mice. Modulation was calculated relative to non-distractor trials within the analysis window (green shaded areas in (**e**)). Right. Corresponding *p*-values (one-tailed bootstrap with an FDR using the Benjamini-Hochberg procedure). Source data are provided as a Source Data file.

population dynamics, are more vulnerable to sensory perturbations in APP-KI mice.

## Disrupted functional interactions across the dorsal cortex in APP-KI mice

We hypothesized that the heightened sensitivity of APP-KI neural activity to sensory perturbations arises from compromised functional interactions across cortical regions. To test this, we first evaluated functional connectivity during inter-trial intervals (ITIs) by computing pairwise correlations of single-neuron activity. To ensure comparable region-by-region activity levels between control and APP-KI mice, we sub-sampled neurons before deriving connectivity measures (Fig. S4a). In both groups, intra-regional connectivity was generally higher than

inter-regional connectivity. However, APP-KI mice exhibited a widespread reduction in functional connectivity relative to controls ($P < 0.001$ for main effects of intra- vs. inter-regions and genotype, two-way ANOVA, Fig. 4a). This reduction persisted even when analysis was restricted to neurons with similar encoding properties (Fig. S4b).

Beyond overall functional connectivity, we also examined trial-by-trial response correlations between neuron pairs. In both control and APP-KI mice, correlations were typically stronger among neurons within the same region. However, these correlations were consistently lower in APP-KI mice across most regions ($P < 0.001$ for main effects of intra- vs. inter-regions and genotype, two-way ANOVA, Fig. 4a).

To further assess inter-regional communication, we applied multivariate linear regression to pairs of cortical regions, measuring how

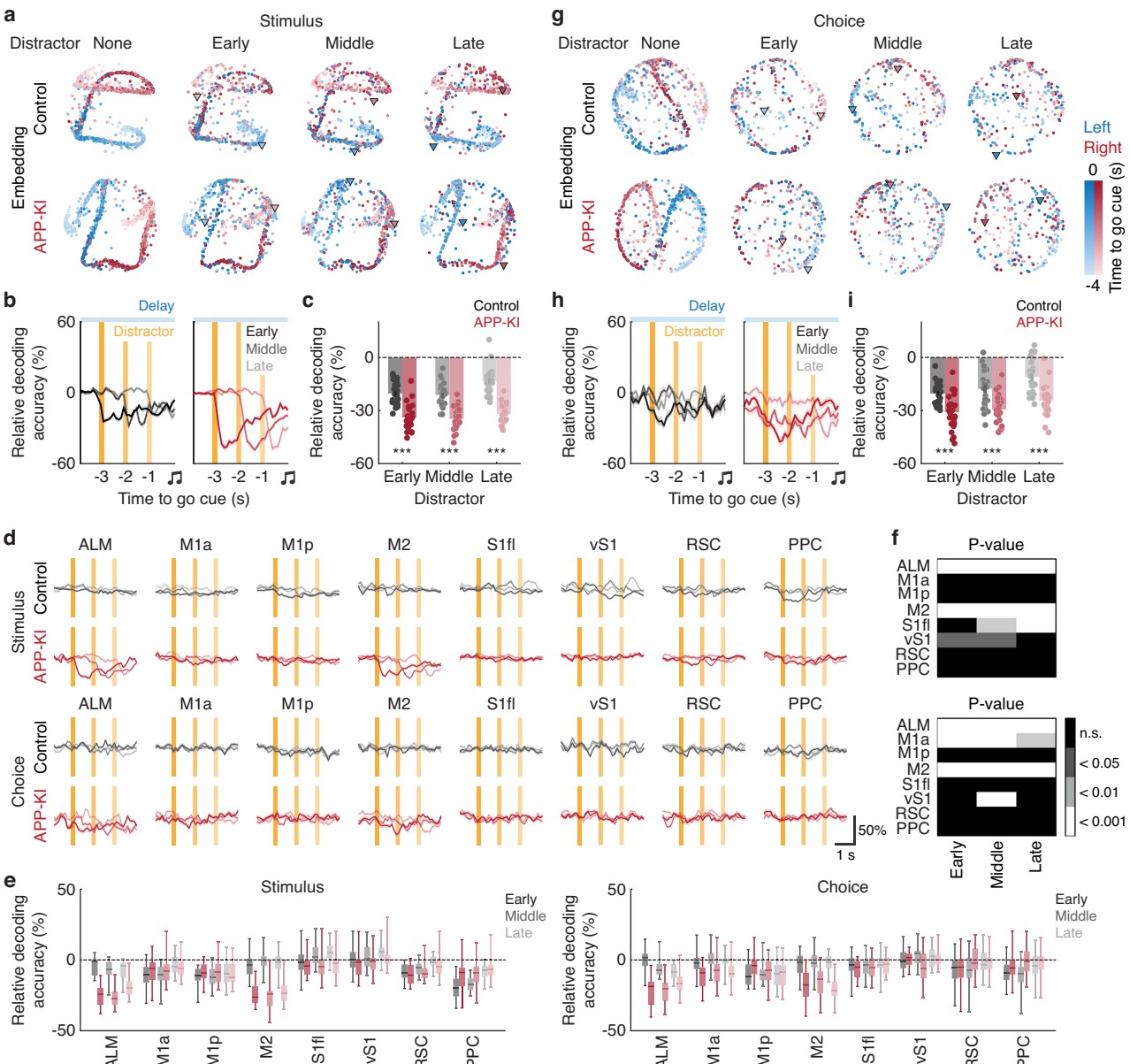

**Fig. 3 | Distractor-mediated reduction of stimulus and choice selectivity in low-dimensional embeddings of neural population activity. a** CEBRA embedding of neural population activity using the stimulus as the behavioral label, visualized across different trial types for control and APP-KI mice. **b** Moment-by-moment decoding accuracy of stimulus selectivity during the delay period in early, middle and late distractor trials for control and APP-KI mice. Relative decoding accuracy was quantified by subtracting non-distractor trial accuracy from distractor trial accuracy. Orange vertical lines indicate the period when the distractor was present. Shaded areas represent mean ± SEM. **c** Summary of the decoding results in (**b**) (***$P < 0.001$ for early, middle and late distractors, $n = 25$ pseudo-mice for control and APP-KI mice, one-tailed bootstrap with an FDR using the Benjamini-Hochberg procedure). Relative decoding accuracy was averaged over a 1-s window following the distractors shown in (**b**). Error bars indicate mean ± SEM. **d** Moment-by-moment decoding accuracy of stimulus and choice selectivity during the delay period in early, middle and late distractor trials for each cortical region in control and APP-KI mice. Relative decoding accuracy was computed using the same procedure as in (**b**). Shaded areas represent mean ± SEM. **e** Summary of the decoding results in (**d**). The analysis window was the same as in (**c**). The boxes denote the 25th and 75th percentiles, the horizontal lines indicate the median and the whiskers represent the minimum and maximum values. **f** Corresponding $p$-values for (**e**) ($n = 25$ pseudo-mice for control and APP-KI mice, one-tailed bootstrap with an FDR using the Benjamini-Hochberg procedure). Error bars indicate SEM. **g** Same as (**a**), but with choice as the behavioral label. **h** Same as (**b**), but for choice selectivity. Shaded areas represent mean ± SEM. **i** Summary of the decoding results in (**h**) (***$P < 0.001$ for early, middle and late distractors, $n = 25$ pseudo-mice for control and APP-KI mice, one-tailed bootstrap with an FDR using the Benjamini-Hochberg procedure). The analysis was performed using the same procedure as in (**c**). Error bars indicate mean ± SEM. Source data are provided as a Source Data file.

well neural activity in one region predicted activity in another. Prediction performance was higher within regions than across regions but was consistently lower in APP-KI mice ($P < 0.001$ for main effects of intra- vs. inter-regions and genotype, two-way ANOVA, Fig. 4a). When examined by trial epoch, prediction performance was similarly reduced in APP-KI mice during stimulus, delay or action periods

($P < 0.001$ for main effects of intra- vs. inter-regions and genotype in all epochs, two-way ANOVA, Fig. S4c).

All three analyses provided convergent evidence of disrupted functional communication in the dorsal cortex of APP-KI mice. Correlations between methods were strong (functional connectivity vs. trial-by-trial correlation: $R^2 = 0.49$, $P = 4.37 \times 10^{-20}$; functional

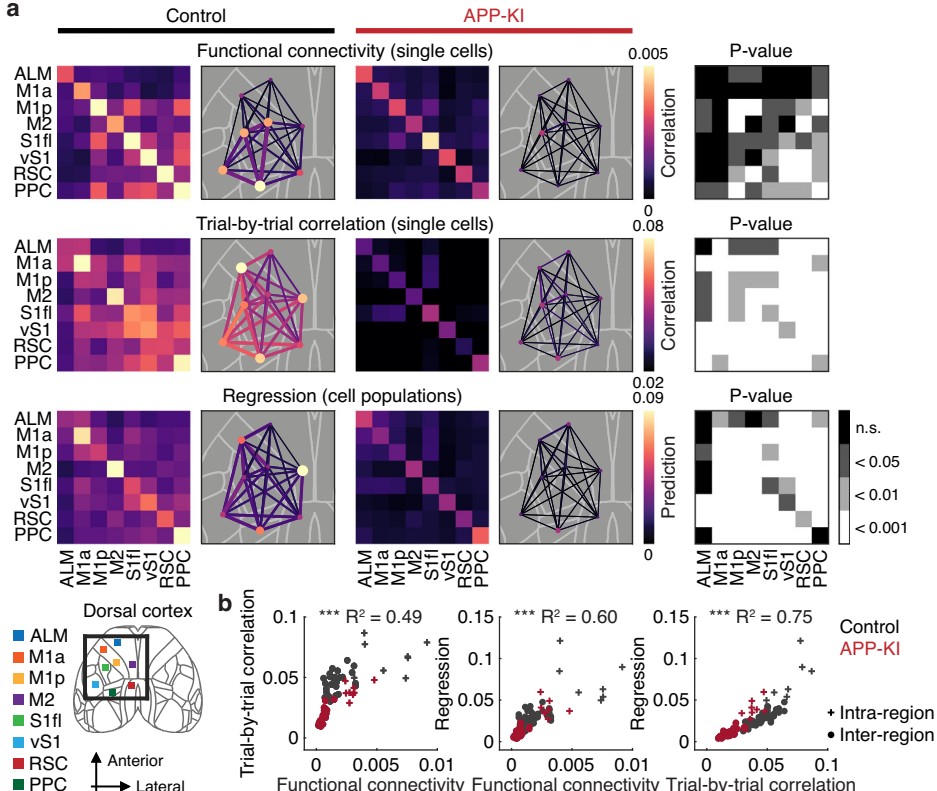

**Fig. 4 | Impaired intra- and inter-regional communication in APP-KI mice.**
**a** Top. Regional communication assessed by functional connectivity between pairs of neurons during inter-trial intervals in control and APP-KI mice. Dot size and color represent intra-regional Pearson correlation coefficients. Line width and color represent inter-regional Pearson correlation coefficients. Cortical region locations are shown for reference beside (**b**). Corresponding $p$-values were determined by a one-tailed bootstrap with an FDR using the Benjamini-Hochberg procedure. Middle. Regional communication evaluated by trial-by-trial correlations in the neural activity of pairs of neurons in control and APP-KI mice. $P$-values were determined as in the top panel. Bottom. Regional communication assessed by linear regression of population activity in control and APP-KI mice. $P$-values were determined as in the top panel. **b** Consistency of metrics used to evaluate functional interactions across the dorsal cortex (functional connectivity vs. trial-by-trial correlation: $R^2 = 0.49$, ***$P = 4.37 \times 10^{-20}$; functional connectivity vs. regression: $R^2 = 0.60$, ***$P = 5.93 \times 10^{-27}$; trial-by-trial correlation vs. regression: $R^2 = 0.75$, ***$P = 1.48 \times 10^{-39}$, $n = 128$ region pairs, Pearson correlation, two-tailed). Source data are provided as a Source Data file.

connectivity vs. regression: $R^2 = 0.60$, $P = 5.93 \times 10^{-27}$; trial-by-trial correlation vs. regression: $R^2 = 0.75$, $P = 1.48 \times 10^{-39}$, $n = 128$ region pairs, Pearson correlation, Fig. 4b). Together, these results indicate a profound impairment in the functional architecture that supports persistent neural activity within the dorsal cortex of APP-KI mice.

## Unstable attractor dynamics in APP-KI mice

The increased susceptibility of APP-KI mice to sensory perturbations may reflect unstable neural network dynamics, particularly those governing choice selection in two-alternative forced-choice tasks. In such tasks, population activity is thought to settle into discrete attractor states corresponding to each choice, with a saddle point separating the respective attractor basins[46,47] (Fig. 5a). To model cortical activity in control and APP-KI mice, we constructed recurrent neural networks (RNNs) using the first-order reduced and controlled error (FORCE) algorithm[48,49] (Fig. 5b). Each unit in the RNN was trained to reproduce the activity of a single neuron, averaged across correct right- or left-choice trials during expert sessions (Fig. 5c and S5a). We then separated population activity related to upcoming choice ("choice axis") from that related to the sensory stimulus by projecting overall activity onto axes that maximized choice selectivity during the pre-action period (1 s before the go cue, lasting 1 s)[44,47].

To assess network stability, we introduced perturbations for 500 ms during the middle of the delay period (Fig. 5b). As predicted, population activity along the choice axis was significantly less stable in APP-KI networks ($P < 0.001$, $n = 30$ RNNs, one-tailed bootstrap, Fig. 5d,

e), consistent with behavioral evidence that APP-KI mice are more prone to sensory perturbations (Fig. 2b). This instability was accompanied by reduced recurrent synaptic strength in APP-KI networks compared with controls ($P < 0.001$, $P < 0.05$, $P < 0.001$ for overall, excitatory and inhibitory strength, respectively, $n = 30$ RNNs, one-tailed bootstrap, Fig. S5b).

To investigate the mechanisms underlying these unstable attractor dynamics, we simulated the reduced functional connectivity observed in APP-KI mice by systematically ablating connections in control RNNs. As the fraction of ablated connections increased, the probability of trial switching also rose ($P = 1.5 \times 10^{-12}$, $n = 30$ RNNs, one-way repeated measures ANOVA, Fig. 5f, g). This effect was consistent across different window sizes of the pre-action period and across sample sizes (Fig. S5c, d). Furthermore, consistent with the CEBRA results (Fig. 3d–f), region-by-region ablations revealed greater vulnerability in areas such as ALM and M2 in APP-KI RNNs (Fig. S5e). Together, these findings indicate that reduced functional connectivity drives unstable network dynamics underlying short-term memory and contributes to the heightened susceptibility to sensory perturbations observed in APP-KI mice (Fig. 5h).

## Reduced spatial degeneracy in cortico-cortical communication in APP-KI mice

Cortical areas interact by selectively routing a subset of their population activity, referred to as a "communication subspace", to downstream regions on a moment-by-moment basis[30,32,33]. Given the

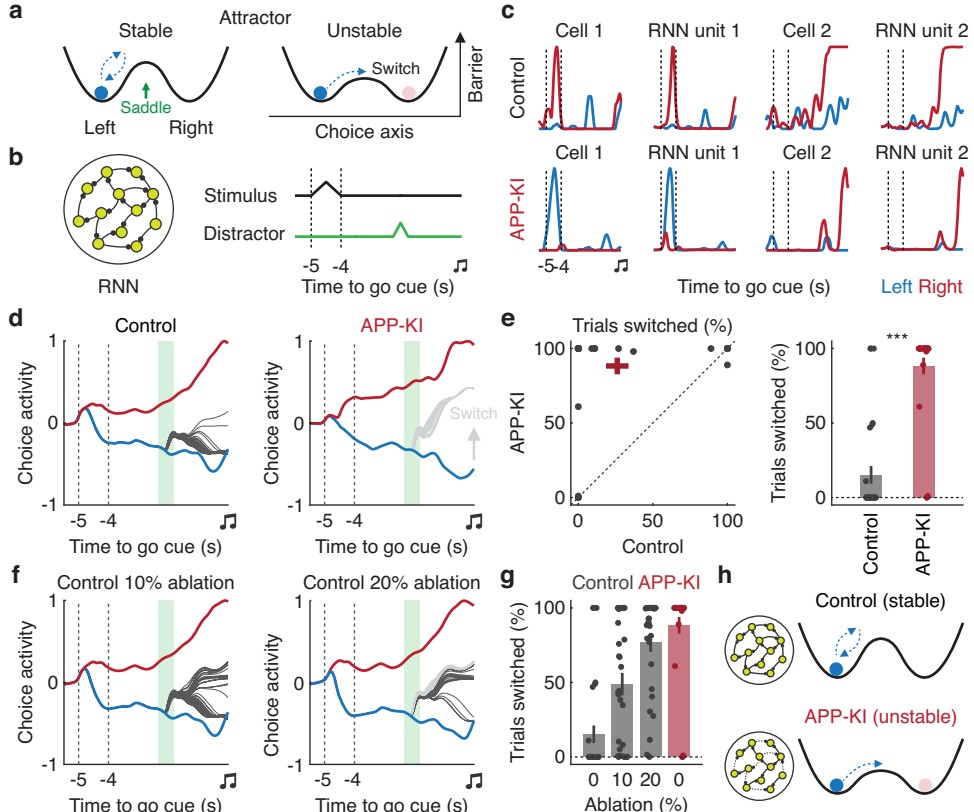

**Fig. 5 | Reduced robustness of attractor dynamics in APP-KI mice. a** Schematic of attractor dynamics along the choice axis during the delay epoch. **b** Schematic of the RNN architecture trained to replicate cortical neural activity. A stimulus pulse was delivered to the RNN to instruct the right trial type. In perturbation trials, a distractor was applied to the trained RNN. **c** Example RNN units (right) reproducing the neural activity of cortical neurons (left) from control and APP-KI mice. **d** Example population activity of RNN units projected onto the choice axis. Blue and red trajectories represent the average choice activity for left and right non-distractor trials, respectively. Gray and black trajectories represent individual perturbation trials where decisions did and did not switch, respectively. Shaded green rectangle denotes the distraction period. **e** Left. Proportions of trials in which

perturbations switched the RNN's choice for control and APP-KI. Red cross indicates mean ± SEM. Right. Mean proportions from the left panel (***$P < 0.001$, $n = 30$ RNNs, one-tailed bootstrap). Error bars indicate mean ± SEM. **f** Example population activity of RNN units projected onto the choice axis with 10% or 20% of connectivity ablated. Trajectory colors and shaded green rectangle are the same as in (**d**). **g** Quantification of network stability as a function of connectivity ablation for control ($P = 1.5 \times 10^{-12}$, F(3, 87) = 27.34, $n = 30$ RNNs, one-way repeated measures ANOVA). Error bars indicate mean ± SEM. **h** Schematic summarizing the result, illustrating how reduced synaptic connectivity leads to unstable attractor dynamics. Source data are provided as a Source Data file.

compromised functional connectivity observed in APP-KI mice, we hypothesized that this subspace structure might also be altered in their dorsal cortex. To test this, we applied multivariate reduced-rank regression to identify the dimensions within a source region's activity that best predicted activity in a target region (Fig. 6a). Consistent with previous work[33], intra-regional interactions required more dimensions than inter-regional interactions to explain downstream neural activity ($P < 0.001$ for main effect of intra- vs. inter-regions and genotype, two-way ANOVA, Fig. S6a, b). Notably, for inter-regional interactions, APP-KI mice required higher-dimensional subspaces than controls, whereas intra-regional dimensionality did not differ significantly between groups (intra-regional difference: $P = 0.09$; inter-regional difference: $P < 0.001$, two-way ANOVA with Tukey-Kramer post-hoc test, Fig. S6a, b). These results suggest that target regions in APP-KI mice rely on a broader set of source activity patterns, reflecting more dispersed information transfer across cortical areas.

The heightened vulnerability of APP-KI mice to distractors suggests reduced robustness in their neural network dynamics. One potential contributor to this vulnerability is diminished degeneracy in inter-regional communication[37]. Degeneracy describes the capacity of distinct elements within a system to perform the same function, thereby providing functional robustness[35]. In particular, spatial degeneracy refers to the presence of multiple parallel pathways

capable of performing the same function, providing flexibility and resilience when one pathway is compromised.

To determine whether APP-KI mice show reduced spatial degeneracy, we examined how similar neural activity patterns within a source region's communication subspace are routed to different downstream target regions (Fig. 6a). High spatial degeneracy arises when similar activity fluctuations are transmitted to multiple targets, ensuring robust sensorimotor transformations even when one pathway is perturbed (Fig. 6b). We hypothesized that reduced spatial degeneracy in APP-KI mice, leading to compromised compensatory mechanisms, underlies their short-term memory vulnerability. To test this, we quantified subspace similarity by measuring the angle between subspaces for different target regions served by the same source region. High subspace similarity indicates that multiple target regions receive similar activity fluctuations, reflecting high spatial degeneracy. In APP-KI mice, subspace similarity was generally lower than in controls, indicating fewer redundant pathways for information transfer. Significant differences were observed in source regions including ALM, M2, vS1, RSC and PPC (main effects: $P = 2.73 \times 10^{-18}$ for genotype and $P = 1.29 \times 10^{-17}$ for source region, two-way ANOVA, ALM: $P < 0.001$; M1a: $P = 0.20$; M1p: $P = 0.90$; M2: $P < 0.01$; S1fl: $P = 1.00$; vS1: $P < 0.01$; RSC: $P < 0.01$; PPC: $P < 0.001$, Tukey-Kramer post-hoc test, Fig. 6c and S6c). Moreover, in APP-KI mice, closer analysis showed that the

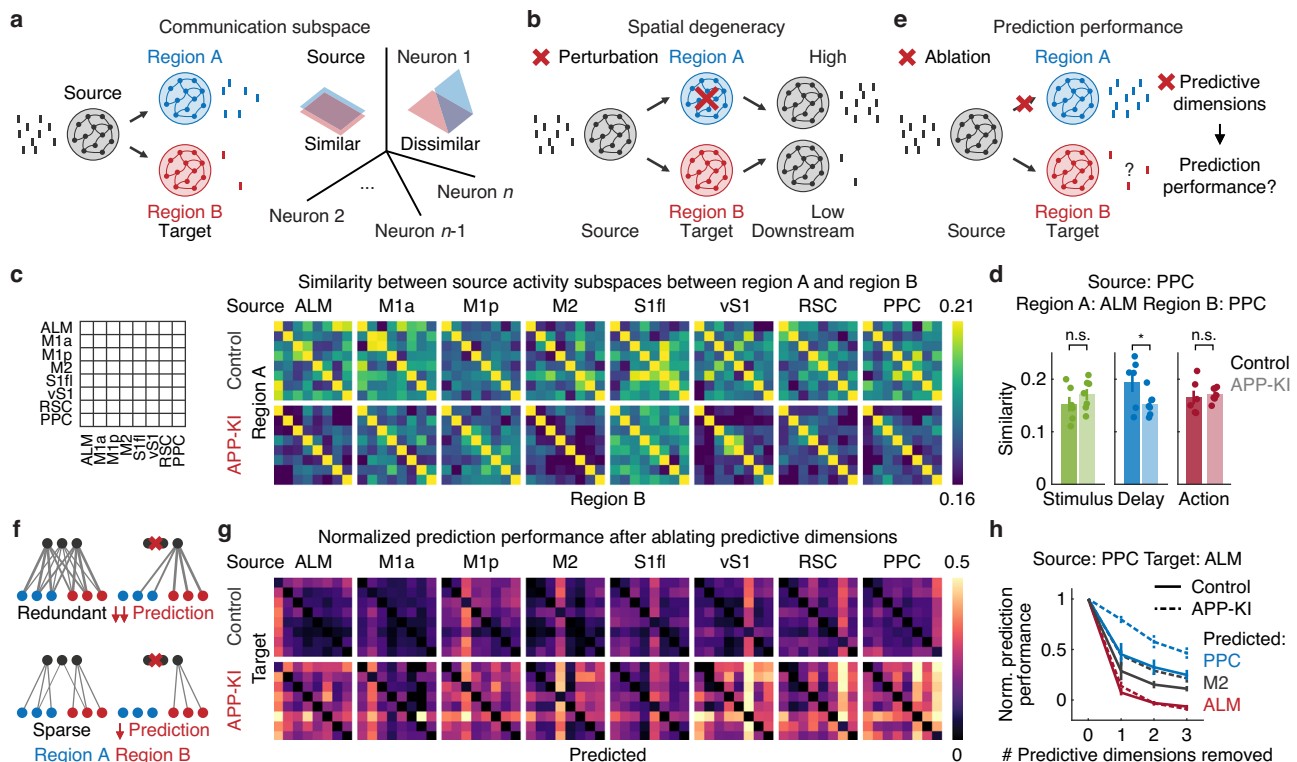

**Fig. 6 | Reduced spatial degeneracy in cortico-cortical communication in APP-KI mice. a** Left. Schematic of low-dimensional interactions between source and target regions (A and B) through communication subspaces. Right. Subspace similarity was computed as the angle between pairs of subspaces. Blue and red parallelograms represent source activity subspace predictive of region A and B, respectively. **b** Schematic illustrating how high vs. low spatial degeneracy influences robustness in neural systems. In high spatial degeneracy (top), disruptions in primary transmission through region A can be compensated by an alternate pathway through region B that conveys the same signal to downstream regions. In low spatial degeneracy (bottom), such compensatory mechanisms are limited, reducing robustness. **c** Left. Schematic illustrating the regions shown in each box of the right panel. Right. Subspace similarity between pairs of target regions for each source region in control and APP-KI mice. Subspace similarity was calculated using the first three dimensions of the communication subspace (main effects: $P = 2.73 \times 10^{-18}$, $F(1, 5125) = 76.65$ for genotype and $P = 1.29 \times 10^{-17}$, $F(7, 5125) = 13.66$ for source region, two-way ANOVA, ALM: $P < 0.001$; M1a: $P = 0.20$; M1p: $P = 0.90$; M2: $P < 0.01$; S1fl: $P = 1.00$; vS1: $P < 0.01$; RSC: $P < 0.01$; PPC: $P < 0.001$, Tukey-Kramer post-hoc test). **d** Subspace similarity between source (PPC) and target (ALM and PPC) regions in control and APP-KI mice during stimulus, delay and action epoch (stimulus: n.s., $P > 0.05$; delay: $*P < 0.05$; action: n.s., $P > 0.05$, $n = 6$ and 7 sessions for control and APP-KI mice, respectively, one-tailed bootstrap with an FDR using the Benjamini-Hochberg procedure). Error bars indicate mean ± SEM. **e** Schematic of ablation of

prediction dimensions between a source region and one target region (region A) and its effect on prediction performance for another target region (region B). **f** Schematic illustrating the relationship between spatial degeneracy and ablation effects. In highly degenerate systems (top), predictive dimensions overlap substantially, so their ablation produces larger reductions in prediction performance for region B in (**e**). In non-degenerate systems (bottom), little overlap exists, and ablation has minimal effect. **g** Normalized prediction performance across source regions in control and APP-KI mice after predictive dimension ablation. Normalized performance was calculated for source-to-predicted regions after removing two predictive dimensions from the communication subspace between specific source-target pairs. When the source and predicted regions were the same (but not the target), the reduction in prediction performance remained relatively unaffected (main effects: $P = 8.11 \times 10^{-7}$, $F(1, 12257) = 24.36$ for source–target identity, $P = 2.47 \times 10^{-306}$, $F(1, 12257) = 1482.79$ for target–predicted identity, $P = 6.52 \times 10^{-123}$, $F(1, 12257) = 568.73$ for source–predicted identity and $P = 3.13 \times 10^{-27}$, $F(1, 12257) = 117.39$ for genotype, four-way ANOVA. **h** Normalized prediction performance of ALM, M2 and PPC when the source and target regions were PPC and ALM, respectively, as a function of the number of predictive dimensions removed, in control and APP-KI mice (Control: PPC-ALM-ALM (source-target-predicted): $n = 7$; PPC-ALM-M2: $n = 7$; PPC-ALM-PPC: $n = 6$ sessions; APP-KI: PPC-ALM-ALM: $n = 13$; PPC-ALM-M2: $n = 10$; PPC-ALM-PPC: $n = 7$ sessions). Error bars indicate mean ± SEM. Source data are provided as a Source Data file.

PPC transmitted less similar activity patterns to ALM and to itself during the delay period (stimulus: n.s., $P > 0.05$; delay: $P < 0.05$; action: $P > 0.05$, $n = 6$ and 7 sessions for control and APP-KI mice, respectively, one-tailed bootstrap with an FDR using the Benjamini-Hochberg procedure, Fig. 6d). Thus, APP-KI mice exhibited reduced spatial degeneracy, with source regions sending dissimilar activity patterns to different target regions.

We further evaluated spatial degeneracy by ablating principal dimensions of a source region's subspace that were predictive of a particular target region's activity. If removing these dimensions also impaired predictions for other targets, it would indicate that many targets share overlapping dimensions (Fig. 6e, f). In both control and APP-KI mice, removing source activity along dimensions that best predicted a given target reduced prediction performance for the same target (Fig. 6g, h). Importantly, prediction performance also declined

when the predicted region differed from the target, suggesting that predictive dimensions in the source region were shared across multiple downstream regions (Fig. 6g, h).

Crucially, this cross-target effect was less pronounced in APP-KI mice (main effects: $P = 8.11 \times 10^{-7}$ for source–target identity, $P = 2.47 \times 10^{-306}$ for target–predicted identity, $P = 6.52 \times 10^{-123}$ for source–predicted identity and $P = 3.13 \times 10^{-27}$ for genotype, four-way ANOVA, Fig. 6g, h), indicating a relative lack of shared predictive dimensions across downstream regions. These effects were robust across different numbers of ablated dimensions (1, 2 or 3, Figs. 6g and S6d). In summary, these findings demonstrate that APP-KI mice exhibit reduced spatial degeneracy in their cortical networks. As a consequence, their cortical processing cannot be compensated by alternative routes, likely contributing to their heightened vulnerability in tasks requiring short-term memory.

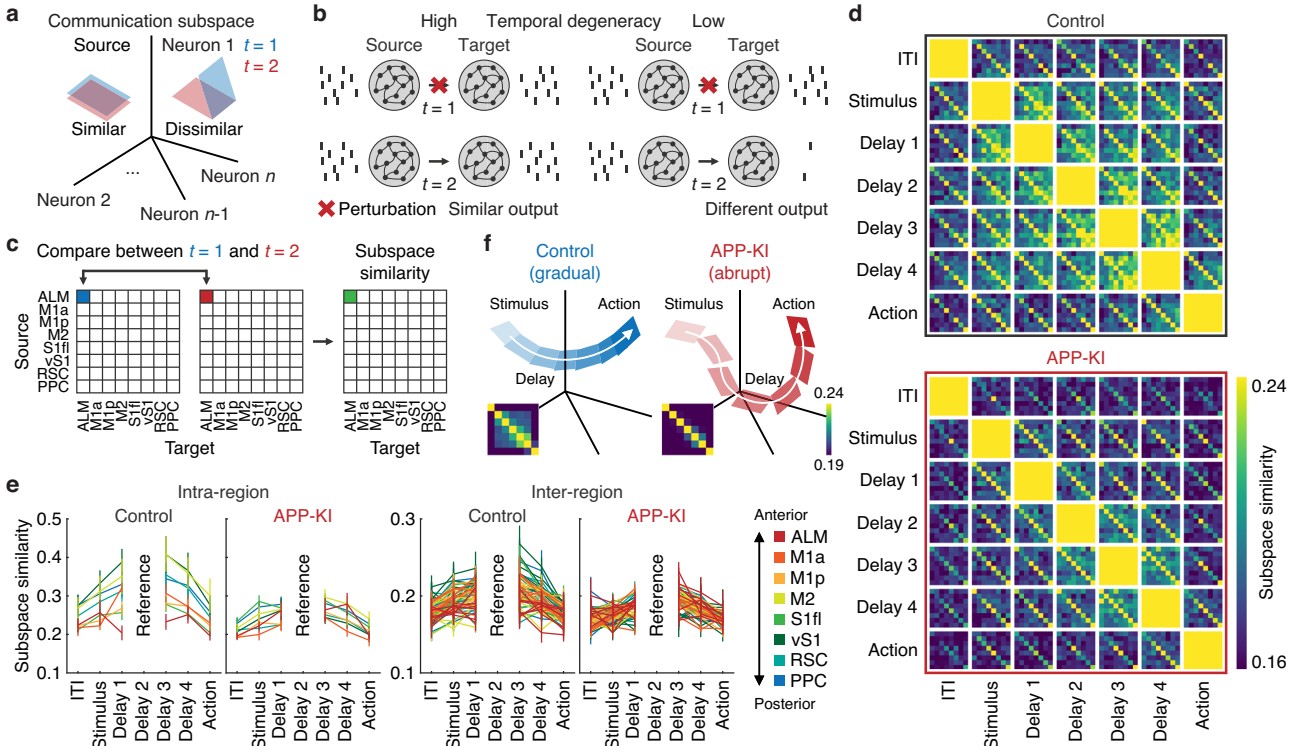

**Fig. 7 | Reduced temporal degeneracy in cortico-cortical communication in APP-KI mice. a** Schematic of high vs. low subspace similarity. Subspace similarity was computed as the angle between pairs of subspaces from different time points. Blue and red parallelograms represent the source activity subspace predictive of a target region at time points t = 1 and t = 2, respectively. **b** Schematic of low-dimensional interactions between source and target regions through communication subspaces at different time points and how high vs. low temporal degeneracy influences the robustness of neural information processing. In systems with high temporal degeneracy, disruption of primary transmission at one time point (t = 1) can be compensated by transmitting the same signal at a later time point (t = 2). In systems with low temporal degeneracy, such compensation does not occur. (**c**). Schematic illustrating how subspace similarity was computed across time. **d** Region-by-region subspace similarity in three-dimensional space evaluated across seven time points (1-s intervals) in control and APP-KI mice. **e** Intra-regional and inter-regional subspace similarity with "Delay 2" as the reference in control and

APP-KI mice. Subspace similarity was higher near the reference time point (main effects: P below machine precision, F(1, 11805) = 2486.02 for intra- vs. inter-regions, $P = 1.07 \times 10^{-143}$, F(5, 11805) = 138.98 for trial epoch, $P = 1.61 \times 10^{-48}$, F(7, 11805) = 34.88 for source region and $P = 9.72 \times 10^{-78}$, F(1, 11805) = 353.55 for genotype, four-way ANOVA). Sample sizes (session numbers) for each source–target pair and genotype are provided in the Source Data file. Different colors represent individual cortical source regions. Error bars indicate mean ± SEM. **f** Schematic summarizing subspace similarity dynamics across trial epochs. Inset shows subspace similarity averaged across region pairs from control and APP-KI mice. Columns (left to right) and rows (top to bottom) correspond to different trial epochs: ITI, stimulus, delay 1, delay 2, delay 3, delay 4 and action. Source subspaces were more similar during the stimulus and delay epochs compared to ITI and action epochs. Control mice exhibited gradual and continuous transitions, whereas APP-KI mice showed more abrupt changes. Source data are provided as a Source Data file.

## Reduced temporal degeneracy in cortico-cortical communication in APP-KI mice

Previous studies have shown that although population activity patterns evolve dynamically during short-term memory, downstream regions can still extract stable stimulus information from a low-dimensional subspace[50,51]. This phenomenon, often referred to as temporal degeneracy, provides redundancy that helps maintain robust information flow despite distractors and noise. Temporal degeneracy allows the same information to be transmitted at different time points, providing a backup mechanism against disruptions. If a signal fails to be routed at one moment, it can be retransmitted later, ensuring that no information is lost (Fig. 7a, b). We therefore reasoned that the vulnerability of short-term memory in APP-KI mice reflects a reduction in temporal degeneracy; as the system loses this compensatory capacity, it becomes less resilient to perturbations[52].

Similar to spatial degeneracy, we hypothesized that the short-term memory vulnerability of APP-KI mice could be partly explained by reduced temporal degeneracy in cortical activity propagation. High temporal degeneracy would mean that similar activity patterns are transmitted from a source region to a target region across different time points. To test this, we measured subspace similarity over 1-s

intervals during the trial, defined as the angle between activity subspaces at different time points for each region pair (Fig. 7c).

Our analysis revealed several key features of inter-regional communication during sensorimotor processing. First, subspace similarity was higher when the source and target were the same, suggesting that individual cortical areas transmit similar activity patterns over time (Fig. 7d). Second, similarity was highest between adjacent trial epochs, reflecting gradual shifts in activity patterns as trials progressed (Fig. 7d). Third, posterior cortical regions exhibited higher subspace similarity, particularly in control mice (Fig. 7e). Fourth, and most importantly, across all conditions, subspace similarity was consistently lower in APP-KI than in control mice (main effects: P below machine precision for intra- vs. inter-regions, $P = 1.07 \times 10^{-143}$ for trial epoch, $P = 1.61 \times 10^{-48}$ for source region and $P = 9.72 \times 10^{-78}$ for genotype, four-way ANOVA, Fig. 7d). These findings remained robust when varying the number of subspace dimensions considered (Fig. S7a–d). In addition, during the stimulus and delay periods, control mice exhibited gradual, continuous transitions in activity patterns, whereas APP-KI mice showed more abrupt shifts (Fig. 7f). Together, these results suggest that APP-KI mice have reduced temporal degeneracy, with fewer redundant or overlapping activity trajectories supporting the same memory across time. Such abrupt transitions in APP-KI mice

likely contribute to their heightened susceptibility to perturbations and short-term memory deficits.

## Discussion

Diverse animal models of AD have been developed to replicate its key neuropathologies and cognitive impairments. However, the precise neural activity patterns linking these phenotypes remain under-explored. In this study, we employed APP-KI mice, which carry the mutated human $App^{NL-G-F}$ gene without overexpression, to investigate the neural mechanisms underlying AD-associated short-term memory deficits. APP-KI mice accumulate Aβ-containing extracellular plaques, providing a physiologically relevant model to AD pathology.

Using two-photon calcium imaging across eight cortical regions, we found that APP-KI mice were more susceptible to distractors during the maintenance phase of short-term memory. This vulnerability was accompanied by disrupted functional connectivity among neuron pairs, both within and across cortical regions. RNN models recapitulated this enhanced sensitivity, suggesting that reduced functional connectivity may underlie the observed susceptibility to perturbations.

While the functional roles of ALM, M2 and PPC in short-term memory have been extensively examined[23,40,46,53–56], the consequences of their impairments in AD mouse models remain relatively unknown. Here, we provide evidence that these regions are especially vulnerable to perturbations in APP-KI mice. Building on previous findings that PPC supports robust attractor dynamics of ALM neurons for choice representations in non-disease mice[44], it is possible that PPC-ALM communication is critical for maintaining short-term memory.

What cellular mechanisms drive this enhanced vulnerability in APP-KI mice? AD is closely linked to synaptic dysfunction[57], and Aβ is known to impair synaptic function and plasticity[8,58–62]. Moreover, increased astrocytosis in APP-KI mice[8–10] may further contribute to aberrant neurotransmission, potentially via excessive GABA release or hydrogen peroxide production by reactive astrocytes[63,64]. Such changes could compromise both anatomical and functional connectivity[65].

Degeneracy, defined as the presence of multiple systems capable of performing the same function, confers functional adaptability and robustness against perturbations[35,37]. With degeneracy, the system can flexibly recruit alternative compensatory strategies when a given pathway is impaired. We reasoned that the short-term memory vulnerability observed in APP-KI mice reflects reduced degeneracy; as the system loses its compensatory capacity, it becomes less resilient to malfunction[52]. Our analysis of communication subspaces revealed that APP-KI mice exhibit reduced spatiotemporal degeneracy compared with controls, indicating that neural circuits critical for short-term memory are more prone to instability. This reduction in degeneracy may stem from synaptic failure, including deficits in long-term potentiation, reduced functional connectivity and inefficiencies in synaptic activity propagation[61,66,67]. Future studies are warranted to establish the causal link between synaptic deficits, reduced degeneracy and their impact on short-term memory.

In summary, our experimental and theoretical findings highlight reduced degeneracy in cortical networks of APP-KI mice as a potential mechanism underlying impaired short-term memory function.

## Methods
### Animals
All procedures were conducted in accordance with guidelines approved by the Institutional Animal Care and Use Committee at Nanyang Technological University (protocol number: A22031). Transgenic mice (*Mus musculus*) used in this study were Thy1-GCaMP6f (GP5.17: The Jackson Laboratory, 025393)[68] and APP-KI ($App^{NL-G-F/NL-G-F}$) mice (RIKEN, RBRC06344). In APP-KI mice, the mutated human $App^{NL-G-F}$ gene (carrying the Swedish, Beyreuther/Iberian and Arctic mutations) was inserted into the endogenous mouse *App* locus[8]. For

this study, unless noted otherwise, Thy1-GCaMP6f mice were referred to as control mice, whereas Thy1-GCaMP6f (hemizygous) × APP-KI (homozygous) mice were referred to as APP-KI mice. All animals (Table S1) were housed in standard cages under controlled conditions (~21 °C and ~62% humidity) with a reversed 12 h:12 h light-dark cycle. Experiments were typically performed during the dark phase. Both male and female mice were included. Sample sizes were determined based on previous studies[47,69].

### Surgery
Adult mice (between 4 and 8 months old) were anesthetized with 1–2% isoflurane. After removing a section of scalp, the exposed bone was cleaned with a razor blade. Using a dental drill, a ~7 mm-wide circular craniotomy was then performed in the left hemisphere, centered 0.20 mm anterior and 1.50 mm lateral to bregma. An imaging window was constructed from a small glass plug (~6 mm in diameter, #2 thickness, Fisher Scientific, 12-540-B) attached to a larger glass base (~8 mm in diameter, #1 thickness, Fisher Scientific, 12-545-D) using ultraviolet-curing adhesive (Norland, NOA 61). The window was placed into the craniotomy, and the gap between the skull and window was filled with 1.5% agarose (Sigma-Aldrich, A9793-50G). A custom-built titanium head-plate was secured to the skull with cyanoacrylate glue and black dental acrylic (Lang Dental, 1520BLK or 1530BLK). Mice received subcutaneous injections of buprenorphine (0.05–0.1 mg/kg of body weight), Baytril (10 mg/kg of body weight) and dexamethasone (2 mg/kg of body weight) and were monitored until full recovery from anesthesia.

### Behavior
Water-deprived control and APP-KI mice were trained on a delayed-response task, initially without distractors, with one session per day. Each trial began with a 1-s tactile stimulus (~20 Hz sweeping) delivered to either the left or right whiskers, followed by a 2-s delay, a 4-s response epoch and an inter-trial interval (ITI, randomly set at 6, 7, or 8 s). The response epoch was initiated by a 4 kHz tone (go cue) lasting 0.2 s, after which both left and right water spouts were moved toward the mouse. Mice were trained to lick the spout corresponding to the side of the tactile stimulus to receive 4 µl of sucrose water (10–15%) as a reward per trial. Incorrect or no responses were punished with white noise and a 6–8-s timeout. To prevent side bias, if mice licked the same spout for more than five consecutive trials, the next trial was forced to be of the opposite type. Each session consisted of 180 trials. Once mice achieved a success rate of at least 75% in one or more sessions, they proceeded to the test phase, during which behavioral performance and neural activity were recorded.

In these test sessions, the delay epoch was extended to 4 s and distracting stimuli were introduced. Distractors were brief tactile stimuli (~20 Hz sweeping, 0.2 s duration) delivered simultaneously to both sides of the snout at one of three time points during the delay (early: 1 s, middle: 2 s, late: 3 s after delay onset) each accounting for ~10% of all trials.

The task structure was controlled using Bpod (Sanworks) with custom MATLAB code, and task variables were recorded with Wavesurfer (Janelia Research Campus) at a 20 kHz sampling rate.

### Two-photon calcium imaging
Calcium imaging was performed in mice aged 6–10 months using a two-photon random access mesoscope (2p-RAM, Thorlabs)[39]. The system was controlled with ScanImage (Vidrio Technologies) and a mode-locked laser (InSight X3, Spectra-Physics) was tuned to 940 nm, delivering ~40 mW at the objective lens. Imaging was conducted at ~9.35 Hz with a spatial resolution of 1 × 0.4 pixel/µm. Eight fields of view (FOVs, 500 × 500 µm each) were simultaneously imaged at depths ranging from ~150 to 200 µm. The stereotaxic coordinates of the imaged regions, relative to bregma, were: ALM: 2.25 mm AP,

1.65 mm ML; M1a: 1.65 mm AP, 2.75 mm ML; M1p: 0.65 mm AP, 1.75 mm ML; M2: 0.50 mm AP, 0.45 mm ML; S1fl: 0.25 mm AP, 2.65 mm ML; vS1: −1.15 mm AP, 3.45 mm ML; RSC: −1.25 mm AP, 0.55 mm ML; PPC: −1.75 mm AP, 2.05 mm ML. Imaging frames were recorded with Wavesurfer and aligned offline to behavior events.

## Amyloid plaque staining

Approximately 9-month-old wild-type and APP-KI mice (without Thy1-GCaMP6f) were deeply anesthetized with ketamine/xylazine (ketamine: 100 mg/kg; xylazine: 8 mg/kg body weight) and transcardially perfused with saline followed by 4% paraformaldehyde (PFA, Sigma Aldrich, 158127-500 G). Brains were extracted, fixed overnight in 4% PFA and cryoprotected in 30% sucrose (Bio Basic, SB0498) at 4 °C for ~48 h. Coronal sections (40 μm) were cut using a cryostat (Leica, CM1950) and stored in 1× phosphate-buffered saline (PBS) at 4 °C. Sections were stained with 1-Fluoro-2,5-bis [(E)−3-carboxy-4-hydroxystyryl] benzene (FSB, 1% stock solution, Sigma Aldrich, 07602), diluted 1:1 in 50% ethanol, followed by incubation in PBS containing 0.05% Triton X-100 (PBST, Sigma Aldrich, T8787, 1:2000) for 30 min at room temperature. After three washes in PBST (15 min each), sections were mounted in polyvinyl alcohol mounting medium (Sigma Aldrich, 10981) and imaged with a slide scanner fluorescence microscope (Zeiss, Axio Scan Z1) at 10× magnification.

## Analysis

**Behavior.** Task performance was calculated separately for each distractor condition (none, early, middle, late) and each session by dividing the number of correct responses by the total number of trials in which mice responded by directional licking. Learning rates were quantified by fitting each mouse's behavioral performance across sessions with a logistic function:

$$y(t) = \frac{L}{1 + e^{-k(t-t_0)}}, \tag{1}$$

where $t$ is the session index, $L$ is the asymptotic performance, $k$ is the learning rate and $t_0$ is the inflection point. Fits were performed using nonlinear least squares using the "lsqcurvefit" function in MATLAB, and the slope $k$ was used as a measure of learning rate.

**Pre-processing of imaging data.** Imaging data from each cortical region were pre-processed independently for motion correction, cell identification and signal extraction using Suite2p[70]. In control mice, the mean ± SEM counts of Thy1-GCaMP−positive cells between 6 and 10 months of age per imaging session were: ALM: 179.71 ± 18.97 ($n = 28$ sessions); M1a: 203.63 ± 16.73 ($n = 32$ sessions); M1p: 361.63 ± 29.05 ($n = 32$ sessions); M2: 193.97 ± 17.89 ($n = 32$ sessions); S1fl: 284.44 ± 23.27 ($n = 32$ sessions); vS1: 140.38 ± 18.26 ($n = 32$ sessions); RSC: 235.94 ± 18.52 ($n = 32$ sessions); and PPC: 239.47 ± 20.54 ($n = 32$ sessions). In APP-KI mice, the corresponding values were: ALM: 192.08 ± 29.55 ($n = 36$ sessions); M1a: 294.11 ± 21.73 ($n = 37$ sessions); M1p: 348.31 ± 37.84 ($n = 36$ sessions); M2: 253.97 ± 23.48 ($n = 34$ sessions); S1fl: 311.72 ± 22.85 ($n = 36$ sessions); vS1: 183.46 ± 23.18 ($n = 39$ sessions); RSC: 314.69 ± 23.20 ($n = 32$ sessions); and PPC: 341.61 ± 20.44 ($n = 38$ sessions). Significant differences between control and APP-KI mice were detected in M1a and RSC ($P < 0.05$) and in PPC ($P < 0.01$, two-tailed Wilcoxon rank-sum test with an FDR using the Benjamini-Hochberg procedure).

Neurons were included for analysis only if their deconvolved z-scored activity exceeded 10 at least once every 10 min (control mice: ALM: 1395; M1a: 2186; M1p: 7593; M2: 1876; S1fl: 3916; vS1: 1847; RSC: 3582; PPC: 3886 neurons from 32 sessions, 7 mice; APP-KI mice: ALM: 2586; M1a: 4988; M1p: 9478; M2: 4018; S1fl: 6922; vS1: 3554; RSC: 6650; PPC: 7854 neurons from 40 sessions, 11 mice). All neural activity was z-scored prior to further analysis to normalize across cells.

**Task-variable selectivity in single neurons.** For each neuron, average activity was calculated within each task epoch (stimulus, delay, early delay, late delay and action) using correct trials. A neuron was classified as selective if its activity differed significantly between left and right trials ($P < 0.01$) with two-tailed Wilcoxon rank-sum test. Within each session, cortical regions with fewer than 10 recorded neurons were excluded from the analysis. Fractions of selective neurons were then computed for each task epoch (stimulus-, delay- or action-selective) across all neurons in control and APP-KI mice.

**Trial-type selectivity in single neurons.** In each cortical region, the activity of left- or right-preferring delay neurons was extracted from left and right correct non-distractor trials, concatenated and normalized such that each neuron's maximum and minimum activity values were set to 1 and 0, respectively. Neurons were then sorted within each region according to the timing of their activity peak.

To compute trial-type selectivity profiles of delay neurons, task-related activity during left non-distractor trials was subtracted from that during right non-distractor trials, separately for correct and incorrect trials. The resulting response profiles were smoothed with a 0.5-s window, yielding upward and downward deflections for right- and left-preferring delay neurons, respectively. The trial-type selectivity profile was then displayed as the difference between these two response profiles. For each neuron, mean activity during the delay epoch was computed and subsequently averaged across neurons within each region.

To evaluate distractor-mediated modulations of trial-type selectivity, correct and incorrect trials were combined, and response profiles were smoothed over 0.5 s for display. Distractor-mediated modulations of trial-type selectivity were quantified within a 1-s window following each distractor and averaged across time. The same measurements were obtained from non-distractor trials, and these values were subtracted from those of distractor trials. Finally, the difference between control and APP-KI mice was computed.

**CEBRA.** To examine whether distractors influenced neural population dynamics in control and APP-KI mice, we applied CEBRA (Consistent EmBeddings of high-dimensional Recordings using Auxiliary variables)[45] to extract low-dimensional neural embeddings during the delay period. Only active neurons were included in the analysis. To construct "pseudo-mice", random subsets of neurons were sampled from the eight cortical regions. CEBRA models were then trained using either stimulus or choice as behavioral labels or in an unsupervised manner.

**CEBRA stimulus.** For stimulus-related analyses, the delay period was represented using time-stamped labels (0-4 s, in increments of 1/frame rate) together with discrete stimulus labels indicating the tactile stimulus type (left stimulus = 1; right stimulus = 0). These labels, combined with deconvolved calcium signals, were used to train CEBRA models. Region-specific neural populations were constructed by aggregating all active cells from each cortical region across sessions. Neural activity from active cells was aligned to their corresponding trial types and time frames within the delay period. The resulting region-specific population matrix had dimensions $n_i \times m$, where $n_i$ denotes the total number of active cells in region $i$ across sessions, and $m$ denotes the total number of frames. Because some sessions did not contain active cells in every region, region-specific populations were derived from different sessions.

For each region-specific population, 200 cells were randomly sampled and combined to generate a $1600 \times m$ matrix, where 1600 represents the total number of active cells across the eight cortical regions. This population was subdivided into four subsets: non-distractor, early-, middle- and late-distractor trials. From the non-distractor subset, 80% of randomly sampled trials were used for training and 20% for testing. Five-fold cross-validation was applied to

the training set to determine optimal hyperparameters, based on the average InfoNCE loss. Candidate hyperparameters included a learning rate of 0.01, weight decay values of 1e-4, 1e-5 and 1e-6, and 512 hidden units. The number of left- and right-stimulus trials was balanced throughout the process. After identifying the best-performing configuration, the model was re-initialized and trained on the full training set with the following parameters: model architecture = "offset1-model"; time offsets = 1; batch size = None; distance = "cosine"; conditional = "time_delta"; temperature = 1; maximum iterations = 2000; output dimension = 3.

Once trained, the CEBRA model was applied to the non-distractor test set as well as to early-, middle- and late-distractor subsets to generate corresponding three-dimensional embeddings. A k-nearest neighbor (kNN) classifier (cosine distance, $k = 5$) was trained on the neural embeddings to decode left vs. right stimulus type. This classifier was then applied to both test and distractor embeddings to compute decoding accuracy. Relative decoding accuracy for distractor trials was quantified by subtracting the accuracy of the non-distractor test set from that of the corresponding distractor subset.

The entire procedure was repeated 25 or 11 times to generate 25 or 11 "pseudo-mice". For each distractor type in each pseudo-mouse, decoding accuracy was averaged over a 1-s window following distractors and then compared between control and APP-KI mice.

**CEBRA choice.** For the choice analysis, stimulus labels were replaced with choice labels indicating the behavioral choice made during the action epoch (left choice = 1; right choice = 0), while all other labels remained unchanged. Region-specific neural populations were obtained by aggregating all active cells from each cortical region across sessions, with neural activity sorted by choice (left or right licking) rather than stimulus type. To match trial numbers across sessions, 40 left-choice and 40 right-choice trials were randomly sampled from the non-distractor condition, and 4 left-choice and 4 right-choice trials were sampled from each distractor condition (early, middle and late) per session. Sessions with fewer available trials were excluded.

The resulting region-specific population had dimensions $n_i' \times m'$, where $n_i'$ denotes the total number of active cells in region $i$ across all included sessions and $m'$ the total number of frames. From each region-specific population, 145 cells were randomly sampled and combined to form a $1160 \times m'$ matrix. The subsequent procedures were identical to the stimulus analysis, except that the kNN classifier was trained to decode left vs. right choice instead of stimulus type.

**CEBRA unsupervised.** To assess whether the separation of embeddings observed in the stimulus analysis reflected an artifact of label supervision, we trained CEBRA models in a fully unsupervised manner. In this setting, only time-stamped labels were provided to guide embedding generation, without supplying stimulus or choice labels. All other training procedures were identical to those described for the stimulus analysis. Following training, a kNN classifier (cosine distance, $k = 5$) was trained on embeddings from the training set to decode left vs. right stimulus identity. The classifier was then applied to test embeddings to compute decoding accuracy. This procedure was repeated 25 times to generate 25 "pseudo-mice".

**CEBRA ablation.** To evaluate the contribution of individual cortical regions to distractor-related neural dynamics, we performed an ablation analysis. For each region, the activity of its neurons was retained, while the activity of neurons in all other regions was replaced by their own mean activity. This procedure was repeated iteratively for each region, isolating one region at a time. The trained CEBRA model and corresponding kNN decoder were then applied to the ablated populations to compute decoding accuracy across trial types. For each distractor type and cortical region, relative decoding accuracy was calculated by subtracting the non-distractor test set accuracy from the distractor subset accuracy. This analysis provided a measure of the extent to which each cortical region contributed to distractor susceptibility in control and APP-KI mice.

**Functional interactions between cortical regions.** To study functional interactions across cortical regions, we performed three independent analyses in control and APP-KI mice: functional connectivity, trial-by-trial correlation and population regression.

For functional connectivity, neural activity during ITIs (−5 to −1 s relative to trial onset) was extracted and concatenated for each neuron. Only neurons whose z-scored deconvolved calcium signals exceeded 10 at least once every 10 min were included. Cortical regions were analyzed further if they contained at least 10 neurons meeting this criterion. To control for potential differences in overall activity levels between control and APP-KI mice, neurons were subsampled to match activity levels across groups. Specifically, for each cortical region and session, we computed the mean z-scored activity of each neuron across ITI frames, retained only those exceeding a preset activity threshold and performed subsampling accordingly. Functional connectivity between neurons was quantified using Pearson correlation coefficients of ITI activity time-series. Correlation values were averaged across neuron pairs within each region pair and then across sessions. For analyses restricted to neurons sharing similar encoding properties, task-variable-selective neurons (stimulus-, delay- or action-selective) were examined separately. Region-by-region comparisons were excluded if fewer than 10 pairs were available.

To compute trial-by-trial correlations, for each neuron, trial-related activity was extracted from −1 to 7 s relative to trial onset and concatenated across all trials. As in the functional connectivity analysis, neurons were subsampled to match activity levels between control and APP-KI mice. Specifically, for each cortical region and session, the mean z-scored activity of each neuron was calculated across trial frames, and only neurons exceeding a preset activity threshold were retained for subsampling. Pairwise Pearson correlation coefficients were then computed from trial-related activity time-series between neuron pairs. These coefficients were averaged within each region pair and subsequently across sessions.

To examine directed interactions between cortical populations, regression analysis was performed on trial-related activity (−1 to 7 s relative to trial onset). Neural activity was concatenated across all trials for each region, and neurons were subsampled using the same activity threshold described above to match activity levels between control and APP-KI mice. For intra-regional interactions, regions with at least 20 neurons were further analyzed, and 10 neurons were randomly sampled as the source while a held-out 10 neurons were sampled as the target. For inter-regional interactions, regions containing at least 10 neurons were included, and 10 neurons were randomly sampled for each source and target region. Regression analysis followed previously established methods (https://github.com/joao-semedo/communication-subspace)[33] and was repeated 20 times for each sampling iteration. Briefly, a linear model was fit to predict target population activity **Y** from source population activity **X** by computing a coefficient matrix **B** according to:

$$\mathbf{Y} = \mathbf{XB}, \tag{2}$$

where **X** and **Y** are $t \times p$ and $t \times q$ matrices, respectively, with $p = q = 10$ neurons and $t$ the number of imaging frames. Each column of the coefficient matrix **B** linearly combines the activity of source neurons to predict the activity of one target neuron. **B** was obtained using ridge regression, an extension of ordinary least-squares (OLS) that minimizes the squared prediction error according to:

$$\mathbf{B}_{\text{Ridge}} = (\mathbf{X}^{\mathsf{T}}\mathbf{X} + \lambda\mathbf{I})^{-1}\mathbf{X}^{\mathsf{T}}\mathbf{Y}, \tag{3}$$

where $\mathbf{I}$ is a $p \times p$ identity matrix and $\lambda$ a regularization constant used to reduce overfitting, determined by 10-fold cross-validation.

**Recurrent neural network.** To study attractor dynamics during short-term memory, recurrent neural networks (RNNs) were built using the first-order reduced and controlled error (FORCE) algorithm. Units in each RNN were trained to reproduce the activity of neurons recorded in the left hemisphere of control and APP-KI mice ($n = 1024$ neurons from 8 regions). Hyperparameters were optimized to achieve a good fit for both left- and right-trial activity. The network dynamics were modeled by the first-order differential equation as follows:

$$\tau \dot{x}(t) = -x(t) + \mathbf{J} \cdot r(t) + \mathbf{W}_{\text{stimulus}} I_{\text{stimulus}}^k + \mathbf{W}_{\text{cue}} I_{\text{cue}} + \varepsilon_{\text{noise}}(t), \quad (4)$$

where $x(t)$ is the membrane current of the network, $\mathbf{J}$ is the recurrent synaptic weight matrix, $r(t)$ is the activity, $\mathbf{W}_{\text{stimulus}}$ and $\mathbf{W}_{\text{cue}}$ are synaptic weight matrices for the stimulus and go cue inputs, and $I_{\text{stimulus}}^k$ and $I_{\text{cue}}$ are the stimulus and go cue inputs for the $k$th trial. $\mathbf{J}$ was initialized as a square matrix of size $n \times n$, with each element sampled from a normal distribution:

$$\mathbf{J} \sim \mathcal{N}\left(0, \frac{g}{\sqrt{n}}\right), \quad (5)$$

where $g > 1$ allowed the randomly initialized network to generate chaotic spontaneous activity prior to training[48]. We set $g = 1.2$. After training, the initial distribution of $\mathbf{J}$ was effectively replaced by the learned weights. The weight vectors for stimulus and go cue were sampled from normal distributions: $\mathbf{W}_{\text{stimulus}} \sim \mathcal{N}(0, 1)$ and $\mathbf{W}_{\text{cue}} \sim \mathcal{N}(0, 0.1)$. The stimulus input $I_{\text{stimulus}}$ was a triangular pulse spanning -5.0 s to -4.0 s relative to the action, while the go cue input $I_{\text{cue}}$ spanned $-4.1$ s to $-4.0$ s. For right trials, the peak amplitude of $I_{\text{stimulus}}$ was drawn from a normal distribution: $I_{\text{stimulus}} \sim \mathcal{N}(1.0, 0.1)$, while for left trials it was fixed at 0. The peak amplitude of $I_{\text{cue}}$ was fixed at 1.0. The noise variable was added at each time step, drawn from a normal distribution: $\varepsilon_{\text{noise}} \sim \mathcal{N}(0, 0.15)$.

The membrane current $x$ was updated at each timestep $t$ by integrating the differential equation using Euler's method, with a neural time constant $\tau = 10$ ms and an integration time constant $\Delta t = 1$ ms. Neural activity $r$ was then obtained by applying a sigmoidal function to the membrane current according to:

$$r(t) = \frac{1}{1 + e^{-\beta(x(t) - \theta)}}, \quad (6)$$

where $\beta = 1.0$ and $\theta = 3.0$. All parameters were kept identical across networks.

**Training and testing of RNNs.** Each unit in the RNN was trained to reproduce the activity of a single neuron, computed as the trial-averaged activity across correct right- or left-choice trials during expert sessions. The training epoch spanned 5.5 s, beginning 0.5 s before stimulus onset and ending at the delay offset. To construct the learning target, neural activity was first transformed into the membrane current, $f$. Units with normalized activity below $1 \times$ SD at all time steps were excluded. Remaining activity values were clipped between 0 and 5, with slight offsets of -0.01 and +0.01, respectively, and then normalized by a fixed value of 5. From this pool, 1024 neurons (128 per region) were randomly sampled with replacement.

Normalized activity $r$ was then transformed into the target function $f$ using the inverse sigmoidal function:

$$f(t) = \theta + \frac{1}{\beta} \ln\left(\frac{r(t)}{1 - r(t)}\right), \quad (7)$$

with parameters $\beta = 1.0$ and $\theta = 3.0$. To increase temporal resolution, $f$ was up-sampled from 9.35 Hz to 93.5 Hz by linear interpolation and smoothing using a 400-ms boxcar moving window filter.

Prior to training, the inverse correlation matrix of the network was initialized as $\mathbf{P} = \alpha \mathbf{I}$ where $\mathbf{I}$ is the identity matrix and $\alpha = 0.005$. The learning rate was set to $\alpha_{learn} = 0.05$, and all RNNs were trained for 1500 epochs. The pseudocode for training was as follows:

**Algorithm. : First-Order Reduced and Controlled Error (FORCE)**

> Initialize $\mathbf{J}$, $x$, $\mathbf{W}_{\text{stimulus}}$, $\mathbf{W}_{\text{cue}}$, $\mathbf{P}$
> for each training episode do
> alternate trial type $k \in \{right, left\}$
> generate $I_{stimulus}^k$ and $\varepsilon_{noise}$
> for each timestep $t$ do
> # state update
> $z(t) \leftarrow \mathbf{J} \cdot r(t) + \mathbf{W}_{\text{stimulus}} \cdot I_{\text{stimulus}}^k + \mathbf{W}_{\text{cue}} \cdot I_{\text{cue}} + \varepsilon_{\text{noise}}(t)$
> $x(t) \leftarrow x(t-1) + \frac{\Delta t}{\tau} \cdot [-x(t-1) + z(t)]$
> # non-linearity
> $r(t) \leftarrow \phi(x(t))$
> # error
> $e(t) \leftarrow z(t) - f(t)$
> # weight update
> $\Delta \mathbf{J}(t) \leftarrow \left[\frac{e(t)}{1 + r^T(t) \cdot \mathbf{P}(t-1) \cdot r(t)}\right] \otimes [\mathbf{P}(t-1) \cdot r(t)]$
> # apply updates
> $\mathbf{J} \leftarrow \mathbf{J} - \alpha_{\text{learn}} \cdot \Delta \mathbf{J}(t)$
> $\mathbf{P}(t) \leftarrow \mathbf{P}(t-1) - \frac{\mathbf{P}(t-1) \cdot r(t) \cdot r^T(t) \cdot \mathbf{P}(t-1)}{1 + r^T(t) \cdot \mathbf{P}(t-1) \cdot r(t)}$
> end for
> end for

Training was designed to allow pairwise comparisons between RNNs trained with neurons from control and APP-KI mice using the same random seed for weight initialization. The mean squared error (MSE) of each trained RNN was computed by comparing its output to the target activity, averaged across trial time and across neurons.

The trained RNNs were then tasked with generating estimated neural activity in the presence of a 500-ms distractor (mean amplitude = 2.5, SD = 0.25) during the middle delay epoch ($-2.3$ to $-1.8$ s relative to the delay offset). Each RNN was presented with 100 trials of three types: left, right and left with distractor.

To analyze the population activity of 1024 units per RNN, dimensionality was reduced by projecting the population activity onto the choice axis. The choice axis maximally separated activity trajectories between left and right choices during the pre-action epoch (1 s before the "go cue", lasting 1 s in duration)[44,47]. Additional pre-action epochs, starting 0.5 or 1.5 s before the "go cue" with durations of 0.5 or 1.5 s, respectively, were also tested to confirm robustness. The choice mode was computed using the population neural activity as the vector difference between trial-averaged activity in right and left trials during the pre-action epoch:

$$\Delta \bar{\mathbf{r}} = \bar{\mathbf{r}}^R - \bar{\mathbf{r}}^L, \quad (8)$$

where $\bar{\mathbf{r}}^R$ and $\bar{\mathbf{r}}^L$ are trial-averaged activities during the pre-action epoch of right and left trials, respectively. $\Delta\bar{\mathbf{r}}$ is an $n \times 1$ weight vector, where $n$ is the number of RNN units. Positive and negative weights were assigned to units preferring the right and left choices, respectively. $\Delta\bar{\mathbf{r}}$ was normalized by its $l^2$ norm to control for the number of units. Projection of the population activity along the choice axis for each trial $k$, denoted $\mathbf{p}^k$, was computed as:

$$\mathbf{p}^k = \mathbf{X}^k \left(\frac{\Delta \bar{\mathbf{r}}}{\sqrt{\sum_i^n |\Delta \bar{\mathbf{r}}_i|^2}}\right), \quad (9)$$

where $\mathbf{X}^k$ is a $t \times n$ matrix of unit activity over time $t$, and $i$ indexes units. $\mathbf{p}^k$ has dimensions $t \times 1$.

After sorting trials into left, right and left with distractor, baseline activity was subtracted from the trial-averaged projections. Baseline activity was defined as the trial-averaged projection for each trial condition during the ITI epoch (0.5 s in duration). To compare across RNNs, baseline-subtracted activity was normalized by the maximum right trial choice activity. A left trial with distractor was considered to have switched its decision if the RNN's choice activity at the end of the delay period was closer to the right trial choice activity in the absence of distractors, defined as above the midpoint between the right and left trial choice activities.

To assess potential differences in recurrent synaptic strength between control and APP-KI RNNs, the mean absolute value of each RNN's recurrent weight matrix **J** was calculated as a measure of overall synaptic strength. To further separate contributions of excitatory and inhibitory connections, the mean absolute values of positive (excitatory) and negative (inhibitory) weights in **J** were computed for each RNN.

To test whether functional connectivity contributes to the robustness of the choice code, recurrent weights were reduced by 10 or 20% for a randomly selected subset of connections from the recurrent synaptic weight matrix **J** in control RNNs, and the frequency of decision switching was computed. This procedure was repeated 20 times for each RNN.

To assess the importance of individual cortical regions, region-specific ablations were performed in control and APP-KI RNNs. For each region, the strength of both outgoing connections to other regions and within-region connections were reduced by 10 or 20%. The proportion of trials in which the RNN's choice was switched by the distractor were calculated and used to evaluate its contribution to decision robustness. To avoid ceiling effects, we only included RNNs that exhibited less than 100% trial switching prior to ablation ($n = 24$ and 6 for control and APP-KI RNN, respectively).

**Communication subspace.** To test whether the target population activity could be predicted by a subspace of the source population activity, the rank of the coefficient matrix **B** was limited to $m$ using reduced-rank regression (RRR) according to:

$$\mathbf{B}_{RRR} = \mathbf{B}_{Ridge}\mathbf{V}\mathbf{V}^{T}, \tag{10}$$

where **V** is a $q \times m$ matrix derived from the singular value decomposition of the covariance matrix of **X** and **Y**, containing the top $m$ principal components. The predicted target population activity was then obtained according to:

$$\hat{\mathbf{Y}}_{RRR} = \mathbf{X}\mathbf{B}_{RRR} = \mathbf{X}\mathbf{B}_{Ridge}\mathbf{V}\mathbf{V}^{T} = \mathbf{X}\bar{\mathbf{B}}\mathbf{V}^{T}, \tag{11}$$

where

$$\bar{\mathbf{B}} = \mathbf{B}_{Ridge}\mathbf{V}. \tag{12}$$

$\bar{\mathbf{B}}$ is a $p \times m$ matrix whose columns represent the predictive dimensions of the source population activity.

Prediction performance was measured as $1 - cvl$ using the normalized squared error between the test data and predictions across different numbers of dimensions (1-10), where $cvl$ is the mean cross-validation loss across folds. Full prediction performance, used to compute population-level functional interactions between cortical regions, was assessed at dimensionality $m$ of 10.

The optimal dimensionality of the reduced-rank regression model was defined as the smallest number of dimensions achieving prediction performance within one SEM of the peak, as determined by 10-fold cross-validation[33].

To compute subspace similarity, for each sampling iteration of source and target neurons, an orthonormal basis for a predictive subspace was obtained by QR decomposition of $\bar{\mathbf{B}}$. The similarity between subspaces was quantified using MATLAB's "subspace" function, which calculates the principal angle between them. Subspace similarity was reported as the cosine of this angle and was evaluated separately during each trial epoch.

**Ablation of activity along predictive dimensions.** To remove the source population activity along the predictive dimensions, the source activity **X** was projected onto the subspace uncorrelated with those dimensions[33]. **M** was defined as:

$$\mathbf{M} = \bar{\mathbf{B}}^{T}\boldsymbol{\Sigma}, \tag{13}$$

where $\boldsymbol{\Sigma}$ is the covariance matrix of the source activity **X**. Singular value decomposition was performed on **M** ($\mathbf{M} = \mathbf{U}\mathbf{D}\mathbf{V}^{T}$) to obtain **V**. An orthonormal basis **Q** for the uncorrelated subspace was then computed using the last $p - m$ columns of **V**, satisfying $\mathbf{M}\mathbf{Q} = \mathbf{0}$, $\mathbf{Q}^{T}\mathbf{Q} = \mathbf{I}$. The source population activity was projected onto this uncorrelated subspace by:

$$\hat{\mathbf{X}} = \mathbf{X}\mathbf{Q}. \tag{14}$$

Prediction performance was computed between $\hat{\mathbf{X}}$ and **Y** using ridge regression, as described above.

### Statistics and reproducibility

Statistical tests and error bar definitions are described in the corresponding figure legends. No statistical method was used to predetermine sample size. Three mice were excluded from the study due to poor imaging quality. Trials were randomized across left, right, no-distractor and distractor conditions. The investigators were not blinded to allocation during experiments and outcome assessment.

### Reporting summary

Further information on research design is available in the Nature Portfolio Reporting Summary linked to this article.

## Data availability

The processed data are available on Zenodo (https://zenodo.org/records/14551514). Source data are provided with this paper.

## Code availability

The code is archived on Zenodo (https://doi.org/10.5281/zenodo.18182775) and is available at https://github.com/HiroshiMakinoLaboratory/AlzheimerShortTermMemory.

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

## Acknowledgements

We thank K. Chung for assistance with animal husbandry. This work was supported by the Singapore Ministry of Education Academic Research Fund Tier 3 MOE2017-T3-1-002 (H.M.), the Mochida Memorial Foundation for Medical and Pharmaceutical Research (H.M.), the Astellas Foundation for Research on Metabolic Disorders (H.M.), the Ichiro Kanehara Foundation for the Promotion of Medical Sciences and Medical Care (H.M.) and JSPS KAKENHI Grant Numbers JP25H01750, JP25H02511.

## Author contributions

G.X. and H.M. conceived the project. G.X., X.C., C.L., and L.A. performed the experiments. H.M. and C.L. analyzed the data. X.C. constructed and analyzed the RNNs. H.M. and C.L. wrote the manuscript with input from all authors.

## Competing interests

The authors declare no competing interests.
