## [Transparent Peer Review file · Nature Communications]

Vulnerability of short-term memory in a mouse model of Alzheimer's disease

Corresponding Author: Professor Hiroshi Makino

Version 0:

Reviewer comments:

Reviewer #1

(Remarks to the Author)

Li et al. investigated the instability of short-term memory maintenance in AD using an animal model. Additionally, they employed RNN modeling to explore potential neural mechanisms underlying this instability. Overall, the paper is well-written, and the experiments are well-motivated.

From the animal model and experimental analyses standpoint, I only have one suggestion which I think will be important for the authors to address:

What does the CEBRA embedding look like when trained without behavioral labels? Additionally, can stimulus identity be decoded above chance from the embedding? Demonstrating this (maybe as a supplementary figure) would be important to ensure that training with behavioral labels does not artificially introduce the dichotomy observed in Figure 3.

For the RNN modeling part, I have a few suggestions:

First, the modeling approach should be described in greater detail in the main text (not just in the method section) to make it clear that the RNN was trained to reproduce the neural data rather than being trained on a similar task.

In Figure 5, the authors show that APP-KI RNNs are more likely to switch trials after distraction compared to control RNNs. Are there significant differences between these models? For example, how do their recurrent weight matrices compare? Do APP-KI RNNs exhibit overall reduced synaptic strength in excitatory and/or inhibitory connections? Also have the authors tried fixed point analysis on these RNNs to characterize the attractor dynamics?

For the RNN ablation study (Fig. 5D and 5F), were the 10% and 20% randomly selected from 1024 neurons? If so, have the authors tried repeating this by randomly selecting from each of the 8 areas to see if there are any specific areas more robust than others?

Other suggestion:

Could the authors expand the last paragraph of the discussion section by incorporating previous studies that support the reduced degeneracy hypothesis?

(Remarks on code availability)

Reviewer #2

(Remarks to the Author)

This manuscript from the Makino lab applies state-of-the-art 2P imaging across cortical regions in control and APP model mice to probe the neural correlates underlying short-term memories (STM). The imaging technique itself, the use of KI mice (as opposed to overexpression models), and the application of some of the latest computational methods are some of the strengths. While these experiments are a useful contribution to the field, I was surprised by the less-than-ideal quality of data presentation (some fonts are too small), correlation between APP mouse age/plaques are missing, details about

experiments in the Results section and figure legends are too brief to allow any reader to appreciate the complexity of this work, the findings read as a list of individual experiments rather than a coherent story (i.e., links between experiments, hypotheses, and other ideas are too brief and not convincing as is), and many parameters in their analyses need to be justified or tested for robustness.

The study is well-designed, and appropriate controls were used. I have the following suggestions to strengthen this study, which will enhance its' impact in the field (in the order that they appear):

1. While in Fig 1B, the authors claim that baseline performance was comparable between control and APP mice, they should plot performance across sessions to support this idea (on a per mouse basis). Any learning rate differences, between groups?
2. It is not appropriate to state "APP mice between 6 and 10 months of age" were used. For each control and especially APP mouse, please provide the specific ages. This is relevant because plaque deposition and cognitive performance is known to be variable across this timeframe in APP mice.
3. In Fig 1C, the plaque image is not clear. Please show a larger image. Also, an example of plaque deposition at the youngest and oldest ages used in this study is required, along with statistical quantification of each cortical region that was imaged in this study. The authors should note any differences in the plaques relative to the cortical layers imaged in their 2P experiments. This is critical because an informed reader may want to correlate plaque deposition to the effect size of STM defects in APP mice. Please correlate the STM behavior defects reported with plaque deposition in the same APP animals. Related to this, for each behavioral readout, please plot the results and statistics at an individual mouse level.
4. Please quantify Thy1-GCaMP-positive cell numbers at the specific ages of mice used and for each cortical region investigated in this study (with statistics).
5. Line 94, the delay period of 4 sec is quite long to simply average out in their analyses. What would be found if the authors separated early vs. late delay periods?
6. Fig 1E, plot individual mouse data per group to show the heterogeneity in their dataset.
7. Lines 120 and 122, should Fig S1B be cited?
8. Plots such as those in Fig 2E are too small to be evaluated. Please increase the size and scale bar information here and across the paper. Readers shouldn't struggle to analyze their results.
9. Line 123, vS1 is incorrect? Please check.
10. Line 124, for this conclusion, please plot neural results considering behavioral performance (on a per mouse basis). Correlating neural and behavior data is critical to support this claim.
11. All figure legends are too brief. It is impossible to appreciate the experiments fully based this information and the Results text. Add critical details and be thorough in the legends (especially for peer review). Every colored symbol/bar in each panel should be described in the legends, and confirm that each panel is fully explained also in the Results section.
12. Line 143, how was 25 pseudo-mice determined? This is quite different from their 2P experimental N numbers per group. It should be supported by showing that comparable pseudo-mice numbers as the experiments would yield comparable results.
13. Lines 141-144, discuss these data in the context of PPC since PPC was first highlighted in their Results as a candidate region relevant for STM defects in APP mice. Why is PPC not significant here?
14. Fig 3D,E,I,J are too small. Improve.
15. Fig 5C should be done more rigorously. Compare their cell data to the RNN data and plot differences with statistical tests.
16. Line 196, how was 1 sec selected? Compare these results to ones if 0.5 or 1.5-2 sec was used, for robustness.
17. Line 199, how was 30 RNNs selected? A robustness analysis is needed.
18. Paragraph starting on Line 202. Why was 1 RNN selected? A more direct approach would be to use multiple RNNs to mimic the multi-region dataset from their 2P work. Then, for each cortical region, this RNN analysis would be compared to the experimental data (for instance, is ALM/M2 or PPC the major contributors?). Also, the work about reducing functional connectivity is more applicable/insightful in a multi-region RNN model.
19. Line 218, the inter-regional interactions require deeper analyses. Focus on the ones particularly relevant for APP observations and describe the differences in these results between control vs. APP.

20. Line 233, the hypothesis is extremely weak/brief. This is a critical transition point between experiments in their study and should be convincing to the reader. Improve and similarly across the Results section, double check to improve other hypotheses/transitions.
21. Lines 239-241, in addition to listing regions that were significantly different, some insightful deeper observations are needed. Without observations, the impact of these experiments/results are poor.
22. Fig 6E, typo in the panel?
23. Line 247 is too brief. Expand and thoroughly discuss the results, not just a brief/general statement.
24. For Fig S5C (and Fig 6F), the main text does not discuss the 1 vs. 3 predictive dimension removal results at all. This is unacceptable.
25. Fig 6G should also be provided for the removal of 1 or 3 predictive dimensions, to contrast with the current figure panel.
26. Line 263, is there some literature that led to this hypothesis? How did the authors think about this, what papers influenced them? Those should be described and cited, as supporting evidence for this idea.
27. Line 267, 1 sec interval needs a justification and robustness analysis.
28. Lines 269-273, quantify for control vs. APP on a per region basis, and plot the results clearly to support these observations.
29. Fig 7D, grey scale shades are very hard to distinguish. Either use different colors or separate graphs in addition to these to improve clarity for the reader.
30. Fig 7E, another example of a poorly justified and described result. Improve text. Add deeper implications of gradual vs. abrupt for STM function.
31. The Discussion section is unacceptable. There is no deeper contextualization of their results while taking into account basic science findings in the field for the same regions studied in this paper, as well as from the AD literature. Significant effort and improvement is needed.
32. Fig S1B, bottom graphs need additional labeling in the figure itself and the legend, to clearly indicate what each plot refers to. The reader shouldn't need to be guessing or deciding themselves.

(Remarks on code availability)

Reviewer #3

(Remarks to the Author)

In this manuscript, "Vulnerability of short-term memory in a mouse model of Alzheimer's disease", Li et al. seek to determine the aberrant neural mechanisms underlying short-term memory deficits observed in Alzheimer's disease (AD). They hypothesize that this deficits in short-term memory are caused by the decreased stability of their recurrent neural network activity. Recurrent neural network is thought to support the sustained neural activity in response to a brief stimulus which is considered to be the neural correlates of short-term memory. Employing two-photon microscope calcium imaging in a mouse model of AD (APP-KI) during a delayed-response tactile discrimination task, the authors report that the neural selectivity in these mice is more susceptible to sensory distractors at both single-neuron and population levels. They observed reduced functional connectivity across dorsal cortex as well as weak spatiotemporal degeneracy, which they concluded to be the central mechanisms underlying the short-term deficits in these mice. The study addresses an important mechanistic question regarding one of the most prevalent neurodegenerative diseases, and the manuscript is written nicely with clear organization. However, I have several concerns that I would like to see addressed. My major and minor concerns are described below.

Major concerns:

1. The age of the mice used in the study is quite variable ranging from 6 to 10 months. This is of some concern when the study involves a model for a progressive neurodegenerative disease. What is the age for the expected onset of AD phenotype in APP-KI mice? How does it relate to the range of subject age used in the study? Does the A β plaque accumulation similar across these 5 months? If not, did the severity of the reported observations correlate with age or the amount of A β plaques observed in APP-KI mice?
2. Throughout the manuscript, there is very little opportunity to see the raw data. Figure 1D shows some raster plots of task-related activity from sample cells, but it appears that those are only from the wildtype mice. What do the general neuronal activity look like in APP-KI mice? What are the total number of active neurons for each group that were used for the analysis shown in Figure 1E? I think the big elephant in the room that is not addressed in the manuscript is the possibility that neural

activity in APP-KI is reduced in general. If that is the case, then it will contribute as a confound to the subsequent network analysis like connectivity, correlation, and regression. The criterion for including the neurons in analysis provided in the method section is very broad: "Neurons whose z-scored deconvolved calcium signal exceeded 10 at least once every 10 min were included for analysis". In the text, it is stated that "neurons were subsampled to match activity levels between control and APP-KI mice to avoid them serving as a potential confounding variable", but no details as to how this was achieved is provided. Figure 2C shows some activity in control and APP-KI mice, but the activity levels are normalized to each cells own maximum and minimum levels, making it difficult to compare among neurons or across groups. Given the limited number of cells that used as criterion for including each cortical region in the analyses (regions were included if they contained at least 20 neurons for intra-regional and 10 neurons for inter-regional analysis, respectively), the lack of clear assessment of the general activity levels of APP-KI is concerning.

3. Also relating to my concern #2, the hypothesized secession of persistent neural activity as the cause of short-term memory impairment is not clearly demonstrated since the relevant data regarding perturbations by distractors provided in the manuscript have been process to "decoding accuracy". What does data presented in Figure 2C look like during the trials that had early, middle, and late distractors?

4. Figure 3B, D, G, I: Are these results from the pseudo-mice generated with random sampling of subsets of neurons from all 8 cortical regions combined or are they from a specific cortical region? If the former, I am surprised that they seem to capture the perturbation only present in a few (two: ALM and M2) of the eight cortical regions? I would appreciate a little more in-depth explanation of these findings.

5. Figure 4: It seems to this reviewer that all three measurements, functional connectivity, trial-by-trial correlation, regression would be critically impacted by overall activity levels, but that seems to be the one metric that is not clearly described in this manuscript.

6. I was intrigued by the data shown in Figure 5G where the fraction of the trials that switched decisions in 20% ablation of connectivity in the controls was comparable to those in APP-KI. Does this 20% ablation resemble the decreased % of neurons exhibiting epoch-specific selectivity in APP-KI (Figure 1E)?

7. Figure 7, Is the temporal degeneracy the population code correlate of "persistent activity" that the authors hypothesized to support short-term memory?

Minor concerns

1. The immunostaining of A β plaques shown in Figure 1C is too small to see and should be accompanied by the staining in control mice of matched age.

2. Figure S2, decoding accuracy looks better for APP-KI mice compared to the controls in the absence of distractors. Is that the case?

(Remarks on code availability)

Version 1:

Reviewer comments:

Reviewer #1

(Remarks to the Author)

The authors have satisfactorily addressed all of my previous comments and concerns.

(Remarks on code availability)

Reviewer #2

(Remarks to the Author)

The authors did a great job addressing my comments from round 1. Please see below for a few final thoughts:

Previous point 4. Please quantify Thy1-GCaMP-positive cell numbers at the specific ages of mice used and for each cortical region investigated in this study (with statistics). --- While they provided this information in their rebuttal, please confirm that it is included in the revised manuscript somewhere.

Previous point 5. Line 94, the delay period of 4 sec is quite long to simply average out in their analyses. What would be found if the authors separated early vs. late delay periods? --- Their response sounds good, but they need to discuss in the main text what it could mean that vS1 is significant in the initial delay period, along with what appears to be a strong trend in the later delay period too (Fig S1e). Could this difference add to those contributed by changes in PPC?

Previous point 22. Fig 6E, typo in the panel? --- The typo is in the word "Prediction".

(Remarks on code availability)

Reviewer #3

(Remarks to the Author)

This is a revised manuscript entitled "Vulnerability of short-term memory in a mouse model of Alzheimer's disease" in which Li et al., examine the aberrant recurrent neural network mechanism underlying short-term memory deficits observed in AD. In response to the previous round of reviews, the authors addressed most if not all of my comments adequately. I have but one lingering comment on the overall activity levels of the mutant vs control APP mice that were used in the study. While I appreciate the authors' effort to provide the mean activity level distribution histogram for each group, general overlap coefficient is not the appropriate measure in this case. I say this because the histograms provided for "trial" (Figure S4a bottom) seems to reflect my original concern, which is that activity levels of APP mutants may be lower compared to that of the controls. I recommend that authors run a KS Test on these two distributions to address this remaining concern.

(Remarks on code availability)

Reviewer #1 (Remarks to the Author):

Li et al. investigated the instability of short-term memory maintenance in AD using an animal model. Additionally, they employed RNN modeling to explore potential neural mechanisms underlying this instability. Overall, the paper is well-written, and the experiments are well-motivated.

We greatly appreciate the reviewer's constructive feedback and support regarding our manuscript. We have conducted additional analyses on CEBRA embeddings and RNN modeling, and have expanded the discussion in accordance with the reviewer's suggestions. For clarity, all major changes have been highlighted in yellow in the revised manuscript.

From the animal model and experimental analyses standpoint, I only have one suggestion which I think will be important for the authors to address:

What does the CEBRA embedding look like when trained without behavioral labels? Additionally, can stimulus identity be decoded above chance from the embedding? Demonstrating this (maybe as a supplementary figure) would be important to ensure that training with behavioral labels does not artificially introduce the dichotomy observed in Figure 3.

We thank the reviewer for raising these points. In response, we trained CEBRA models on a dataset without behavioral labels and evaluated the trained models on an independent testing dataset. In both the training and testing datasets, the resulting embeddings formed two clearly separable clusters in the latent space, corresponding to left- and right-trial types. This indicates that the dichotomy observed in **Figure 3** is not an artifact of behavioral labels. Moreover, decoding accuracies for stimulus identity from the testing dataset were significantly above chance level (50%). These results are now presented in **Figure S3b**. The corresponding results and methods are described in lines 136–137 (page 5) and lines 794–801 (pages 40) of the revised manuscript.

For the RNN modeling part, I have a few suggestions:

First, the modeling approach should be described in greater detail in the main text (not just in the method section) to make it clear that the RNN was trained to reproduce the neural data rather than being trained on a similar task.

We apologize for the earlier lack of clarity. To address this, we have revised the text in lines 194–196 (page 6) as follows: “Each unit in the RNN was trained to reproduce the activity of a single neuron, averaged across correct right- or left-choice trials during expert sessions (**Figure 5c and S5a**).”

In Figure 5, the authors show that APP-KI RNNs are more likely to switch trials after distraction compared to control RNNs. Are there significant differences between these models? For example, how do their recurrent weight matrices compare? Do APP-KI RNNs exhibit overall reduced synaptic strength in excitatory and/or inhibitory connections?

We thank the reviewer for these insightful comments. To assess potential differences between control and APP-KI RNNs, we calculated the mean absolute value of each RNN’s recurrent weight matrix. APP-KI RNNs exhibited a slight but statistically significant reduction in recurrent synaptic strength compared to control RNNs (control: $0.0614 \pm 1.17 \times 10^{-4}$; APP-KI: $0.0607 \pm 9.24 \times 10^{-5}$, $P < 0.001$, $n = 30$ RNNs per group, one-tailed bootstrap). This reduction was present in both excitatory and inhibitory connections. Specifically, the mean absolute value of positive (excitatory) weights was slightly but significantly lower in APP-KI RNNs (control: $0.0631 \pm 1.17 \times 10^{-4}$; APP-KI: $0.0629 \pm 9.80 \times 10^{-5}$, $P < 0.05$, $n = 30$ RNNs per group, one-tailed bootstrap), while the mean absolute value of negative (inhibitory) weights showed a more pronounced reduction (control: $0.0598 \pm 1.22 \times 10^{-4}$; APP-KI: $0.0584 \pm 9.04 \times 10^{-5}$, $P < 0.001$, $n = 30$ RNNs per group, one-tailed bootstrap). These findings align with our observation of reduced functional connectivity in APP-KI mice. The results are now shown in **Figure S5b**, and the corresponding descriptions and methods are provided in lines 202–205 (pages 6) and lines 928–931 (page 44) of the revised manuscript.

Also have the authors tried fixed point analysis on these RNNs to characterize the attractor dynamics?

We performed fixed-point analysis on control and APP-KI RNNs following prior studies (Finkelstein et al., 2021; Mante et al., 2013) to characterize their attractor dynamics. In both models, we identified stable fixed points corresponding to the left- and right-choice trajectories, consistent with choice-selective attractors. However, the separation between basins was not well defined; rather than a single saddle point dividing the two attractors, we frequently observed multiple saddles or slow unstable points. This phenomenon, also reported in **Extended Data Fig. 6d and g** of (Finkelstein et al., 2021), fragments the basin boundary, making it difficult to define a single decision surface whose position and stability properties reliably reflect attractor robustness. As a result, standard fixed-point metrics (e.g., distance to saddle) did not yield a clear or unique measure of dynamic stability, and we therefore did not rely on fixed-point analysis for stability assessment in our models.

For the RNN ablation study (Fig. 5D and 5F), were the 10% and 20% randomly selected from 1024 neurons?

The 10% and 20% ablations were performed by randomly selecting connections from the recurrent synaptic weight matrix, irrespective of their anatomical origin. In the revised manuscript, details about this procedure can be found in lines 932–935 (page 44).

If so, have the authors tried repeating this by randomly selecting from each of the 8 areas to see if there are any specific areas more robust than others?

We thank the reviewer for this insightful comment. To assess the importance of individual brain regions, we performed region-specific ablations in control and APP-KI RNNs. For each region, we reduced the strength of both outgoing connections to other regions and within-region connections by 10 or 20%. We then quantified the proportion of trials in which the RNN's choice was switched by the distractor. This procedure was repeated separately for each region to evaluate its contribution to decision robustness. To avoid ceiling effects, we focused on those RNNs that exhibited less than 100% trial switching prior to ablation (control RNNs: $n = 24$; APP-KI RNNs: $n = 6$).

In APP-KI RNNs, ALM and M2 showed significantly increased proportions of distractor-induced choice switching compared to no ablation at both 10% and 20% ablation ($P < 0.001$, paired one-tailed bootstrap with an FDR using the Benjamini–Hochberg procedure), whereas other regions exhibited non-significant changes. In control RNNs, significant increases were observed in ALM, M1a, M1p, S1fl, RSC and PPC under the same comparisons ($P < 0.001$, paired one-tailed bootstrap with an FDR using the Benjamini–Hochberg procedure).

These results were generally consistent with **Figure 3e** where ALM and M2 in APP-KI mice were more vulnerable to distractors than the other regions. The results are now shown in **Figure S5e**, and the corresponding descriptions and methods are provided in lines 210–212 (pages 7) and lines 936–941 (page 44–45) of the revised manuscript.

Other suggestion:

Could the authors expand the last paragraph of the discussion section by incorporating previous studies that support the reduced degeneracy hypothesis?

We thank the reviewer for this comment and agree that the original Discussion section required further elaboration. Accordingly, we have expanded the last few paragraphs of the Discussion to incorporate prior studies supporting the reduced degeneracy hypothesis in AD (lines 322–332, page 9–10).

References

Finkelstein, A., Fontolan, L., Economo, M.N., Li, N., Romani, S., and Svoboda, K. (2021). Attractor dynamics gate cortical information flow during decision-making. *Nat Neurosci* 24, 843-850. 10.1038/s41593-021-00840-6.

Mante, V., Sussillo, D., Shenoy, K.V., and Newsome, W.T. (2013). Context-dependent computation by recurrent dynamics in prefrontal cortex. *Nature* 503, 78-84. 10.1038/nature12742.

Reviewer #2 (Remarks to the Author):

This manuscript from the Makino lab applies state-of-the-art 2P imaging across cortical regions in control and APP model mice to probe the neural correlates underlying short-term memories (STM). The imaging technique itself, the use of KI mice (as opposed to overexpression models), and the application of some of the latest computational methods are some of the strengths. While these experiments are a useful contribution to the field, I was surprised by the less-than-ideal quality of data presentation (some fonts are too small), correlation between APP mouse age/plaques are missing, details about experiments in the Results section and figure legends are too brief to allow any reader to appreciate the complexity of this work, the findings read as a list of individual experiments rather than a coherent story (i.e., links between experiments, hypotheses, and other ideas are too brief and not convincing as is), and many parameters in their analyses need to be justified or tested for robustness.

The study is well-designed, and appropriate controls were used. I have the following suggestions to strengthen this study, which will enhance its' impact in the field (in the order that they appear):

We appreciate the reviewer's constructive feedback, which has been extremely helpful in refining our manuscript. In response, we have improved data presentation, incorporated previous findings to illustrate the correlation between APP-KI mouse age and plaque deposition, and expanded both the Results section and figure legends for clarity. We have also revised the manuscript to more clearly connect the experiments to our hypotheses and conducted additional analyses to test the robustness of key parameters. We believe these amendments have substantially improved the manuscript. For clarity, all major changes have been highlighted in yellow in the revised manuscript.

1. While in Fig 1B, the authors claim that baseline performance was comparable between control and APP mice, they should plot performance across sessions to support this idea (on a per mouse basis). Any learning rate differences, between groups?

We plotted the learning curves of individual mice (**Figure S1a**). We compared the number of sessions required to reach the expert level and again found no significant difference between the two groups ($P = 0.98$, $n = 7$ and 11 mice for control and APP-KI mice, respectively, two-tailed Wilcoxon rank-sum test, **Figure S1b**). To quantify learning rates, we fitted each mouse's performance across sessions with a logistic function and extracted the slope. We found no significant difference in learning rates between control and APP-KI mice ($P = 1.0$, $n = 7$ and 11 mice for control and APP-KI mice, respectively, two-tailed Wilcoxon rank-sum test, **Figure S1b**). These results support our conclusion that learning speed is comparable between control and APP-KI mice. We now describe

these results and corresponding methods in line 74 (page 3) and lines 700–705 (page 37–38) of the revised manuscript.

2. It is not appropriate to state “APP mice between 6 and 10 months of age” were used. For each control and especially APP mouse, please provide the specific ages. This is relevant because plaque deposition and cognitive performance is known to be variable across this timeframe in APP mice.

We thank the reviewer for this suggestion. In response, we have provided the exact ages of the individual mice used in the study (**Table S1**).

3. In Fig 1C, the plaque image is not clear. Please show a larger image. Also, an example of plaque deposition at the youngest and oldest ages used in this study is required, along with statistical quantification of each cortical region that was imaged in this study. The authors should note any differences in the plaques relative to the cortical layers imaged in their 2P experiments. This is critical because an informed reader may want to correlate plaque deposition to the effect size of STM defects in APP mice. Please correlate the STM behavior defects reported with plaque deposition in the same APP animals. Related to this, for each behavioral readout, please plot the results and statistics at an individual mouse level.

We thank the reviewer for these thoughtful suggestions. In response, we have replaced the original image in **Figure 1c** with an enlarged version.

We agree that providing examples of plaque deposition at both the youngest and oldest ages examined in this study could help link amyloid pathology to short-term memory impairment. However, prior work (Saito et al., 2014) has shown that in APP-KI mice, A β plaques begin to accumulate as early as 2 months of age and reach near-saturation levels by approximately 7 months. In our study, the youngest APP-KI mouse was 6 months and 25 days old, and the oldest was 9 months and 28 days old. Because this age range falls within the plateau phase of plaque accumulation, we do not expect substantial variability in plaque burden across our APP-KI mice.

To confirm this, we compared task performance between 6-month-old and 9-month-old APP-KI mice and found no significant behavioral differences between the two groups across all conditions (none, early, middle and late distractors) ($p > 0.05$, $n = 9$ and 16 sessions for 6-month-old and for 9-month-old APP-KI mice, respectively, two-tailed Wilcoxon rank-sum test with an FDR using the Benjamini–Hochberg procedure, **Figure S1d**). The absence of behavioral differences suggests comparable plaque burden, and therefore comparable effects on short-term memory, within the age range used in our study. These results are now included in lines 79–81 (page 3) of the revised manuscript.

We fully agree that correlating plaque deposition with short-term memory in the same animals would provide valuable insights into the relationship between A β pathology and cognitive impairment. However, for the

reasons described above, we do not expect sufficient variability in A β deposition to allow for such an analysis. In addition, A β imaging was performed in a separate cohort of mice, since calcium imaging required crossing APP-KI mice with Thy1-GCaMP6f mice for two-photon experiments.

Finally, to address the reviewer's last point, we have plotted each behavioral readout for individual mice (**Figure S1a**).

4. Please quantify Thy1-GCaMP-positive cell numbers at the specific ages of mice used and for each cortical region investigated in this study (with statistics).

We quantified the number of Thy1-GCaMP-positive cells in control and APP-KI mice between 6 and 10 months of age across eight cortical regions. In control mice, the mean \pm SEM cell counts per imaging session were: ALM, 179.71 \pm 18.97 (n = 28); M1a, 203.63 \pm 16.73 (n = 32); M1p, 361.63 \pm 29.05 (n = 32); M2, 193.97 \pm 17.89 (n = 32); S1fl, 284.44 \pm 23.27 (n = 32); vS1, 140.38 \pm 18.26 (n = 32); RSC, 235.94 \pm 18.52 (n = 32); and PPC, 239.47 \pm 20.54 (n = 32). In APP-KI mice, the corresponding values were: ALM, 192.08 \pm 29.55 (n = 36); M1a, 294.11 \pm 21.73 (n = 37); M1p, 348.31 \pm 37.84 (n = 36); M2, 253.97 \pm 23.48 (n = 34); S1fl, 311.72 \pm 22.85 (n = 36); vS1, 183.46 \pm 23.18 (n = 39); RSC, 314.69 \pm 23.20 (n = 32); and PPC, 341.61 \pm 20.44 (n = 38). Significant differences between control and APP-KI mice were detected in M1a and RSC (P < 0.05) and in PPC (P < 0.01, two-tailed Wilcoxon rank-sum test with an FDR using the Benjamini-Hochberg procedure).

Although APP-KI mice exhibited a modest but statistically significant increase in the number of Thy1-GCaMP-positive cells in several regions, we controlled for these differences in downstream analyses to ensure fair group comparisons. Specifically, we randomly subsampled an equal number of cells from each cortical region across groups for both CEBRA-based neural embedding extraction and RNN training. In addition, neurons were subsampled to match overall activity levels between control and APP-KI mice, thereby minimizing potential confounding effects on functional interaction and communication subspace analyses.

To test whether the reduced proportion of PPC neurons with task epoch-specific selectivity in APP-KI mice (**Figure 1e**) was simply due to larger cell counts, we assessed the correlation between the number of cells and the cell fractions in **Figure 1e** for both control and APP-KI mice. In both groups, the results revealed no significant correlation (an FDR using the Benjamini-Hochberg procedure). Therefore, the reduced PPC selectivity observed in APP-KI mice is unlikely to be due to larger cell numbers.

5. Line 94, the delay period of 4 sec is quite long to simply average out in their analyses. What would be found if the authors separated early vs. late delay periods?

We divided the delay period into two epochs: early delay (first 2 seconds) and late delay (last 2 seconds). We then quantified the fraction of cells exhibiting trial-type selectivity separately for each epoch. Overall, the

findings closely resemble those in **Figure 1e**, showing a reduced proportion of neurons with delay selectivity, with the most pronounced differences observed in the PPC. These results are now presented in **Figure S1e** and are described in line 91 (page 3) of the revised manuscript.

6. Fig 1E, plot individual mouse data per group to show the heterogeneity in their dataset.

We have updated **Figure 1e** to display the data from individual mice.

7. Lines 120 and 122, should Fig S1B be cited?

We thank the reviewer for pointing this out. The delay results in the original **Figure S1b** (now **Figure S2b**) indeed support the statements in lines 120 and 122 (now lines 117 and 119 on page 4 of the revised manuscript). We have accordingly revised the text.

8. Plots such as those in Fig 2E are too small to be evaluated. Please increase the size and scale bar information here and across the paper. Readers shouldn't struggle to analyze their results.

We have increased the size of these figures and added scale bar information to improve readability. We have also applied similar amendments to other figures.

9. Line 123, vS1 is incorrect? Please check.

This is correct. The conclusion, “For neurons with stimulus-selective responses, this reduction was most significant in posterior regions such as vS1 and PPC.”, was based on **Figure S2b** (top row, originally **Figure S1b**), which illustrates the difference in region-specific distractor-mediated modulation of trial-type selectivity between control and APP-KI mice. We have described these results in lines 119–122 on page 4.

10. Line 124, for this conclusion, please plot neural results considering behavioral performance (on a per mouse basis). Correlating neural and behavior data is critical to support this claim.

We thank the reviewer for this thoughtful suggestion. In our current analysis, however, trial-type selectivity of stimulus-, delay-, and action-selective neurons was computed by pooling neurons across all mice within each group. We adopted this approach because the number of neurons sampled from each region varied across mice due to practical considerations such as partial window occlusion. As a result, we were unable to directly correlate neural selectivity with behavioral performance on an animal-by-animal basis.

11. All figure legends are too brief. It is impossible to appreciate the experiments fully based this information and the Results text. Add critical details and be thorough in the legends (especially for peer review). Every colored symbol/bar in each panel should be described in the legends, and confirm that each panel is fully explained also in the Results section.

We thank the reviewer for the feedback. While we had initially shortened the figure legends to comply with journal word limits, we have now revised all legends for peer review to include detailed experimental descriptions as well as explanations of colored symbols and bars.

12. Line 143, how was 25 pseudo-mice determined? This is quite different from their 2P experimental N numbers per group. It should be supported by showing that comparable pseudo-mice numbers as the experiments would yield comparable results.

The number of 25 pseudo-mice was determined empirically by testing multiple pseudo-population sizes and evaluating the consistency of key results, including stimulus and choice decoding accuracy as well as low-dimensional neural embeddings. Importantly, the results obtained with 25 pseudo-mice were highly consistent with those using a smaller number (11 pseudo-mice per group), which corresponds to the actual experimental sample size. This consistency across different pseudo-mouse group sizes supports the robustness and generalizability of our findings.

For stimulus selectivity, APP-KI mice exhibited significantly greater sensitivity to distractor perturbations ($P < 0.001$ for early, middle and late distractors, $n = 11$ pseudo-mice per group, one-tailed bootstrap with an FDR via the Benjamini–Hochberg procedure). These results were consistent with those obtained using 25 pseudo-mice. Region-level analyses also replicated the key observations; ALM and M2 consistently contributed most to the heightened sensitivity in APP-KI mice across all distractor conditions, mirroring the findings derived from the 25 pseudo-mice analysis.

For choice selectivity, APP-KI mice similarly demonstrated increased vulnerability to distractors ($P < 0.05$ for early and middle distractors, and $P < 0.001$ for late distractor, $n = 11$ pseudo-mice per group, one-tailed bootstrap with an FDR via the Benjamini–Hochberg procedure), again replicating the findings from the 25 pseudo-mice analysis. At the regional level, M2 consistently exhibited elevated sensitivity across all distractor epochs, while ALM showed pronounced sensitivity during early and middle distractors.

These results are now presented in **Figure S3c-f and h-k**, and are described in lines 144-145 and 154–155 on page 5.

13. Lines 141-144, discuss these data in the context of PPC since PPC was first highlighted in their Results as a candidate region relevant for STM defects in APP mice. Why is PPC not significant here?

We thank the reviewer for raising this point. As correctly noted, in **Figure 1e** PPC was identified as a region with reduced fractions of neurons exhibiting task-epoch-specific selectivity in APP-KI mice. These results were obtained from “no-distractor” trials. In contrast, **Figures 3d–f** show that the PPC of APP-KI mice was not more sensitive to perturbations than that of control mice during “distractor” trials.

In our previous work (Chia et al., 2023), we demonstrated that choice selectivity and its robustness in ALM were enhanced by PPC in a non-disease mouse model. Our current findings suggest that the reduced fraction of PPC neurons discriminating trial types in APP-KI mice leads to unstable stimulus/choice representations in ALM when distractors are present. The direct effect of distractors on PPC itself, however, may be less apparent than on ALM.

14. Fig 3D,E,I,J are too small. Improve.

We have increased the size of these panels (now **Figures 3d and 3e**) to improve clarity.

15. Fig 5C should be done more rigorously. Compare their cell data to the RNN data and plot differences with statistical tests.

In response to the reviewer’s comment, we computed the mean squared error (MSE) for each trained RNN by averaging the difference between its output and the target activity across time within trials and across neurons. Each unit in the RNN was trained to reproduce the activity of a single neuron recorded from either control or APP-KI mice. The mean MSE \pm SEM was 0.016 ± 0.0027 for control and 0.014 ± 0.00041 for APP-KI mice ($P = 0.19$, $n = 30$ RNN models per group, two-tailed Wilcoxon rank-sum test), indicating that the RNN models were trained equally well on neural data from both groups. These results are now presented in **Figure S5a**.

16. Line 196, how was 1 sec selected? Compare these results to ones if 0.5 or 1.5-2 sec was used, for robustness.

To assess the impact of distractors on choice-related activity, we computed the choice mode during the last 1 second before the “go cue”. This window is approximately equidistant from the distractor offset and the “go cue”, making it well-suited to isolate sustained effects of the distractor. A shorter window (e.g., last 0.5 seconds) would predominantly capture motor preparation, as animals (and trained RNNs) begin initiating responses in this period. Conversely, a longer window (e.g., last 1.5 seconds) would place the analysis closer to the distractor and risk capturing early sensory components. Thus, the 1-second window minimizes confounds from both early

and late phases and provides a more reliable measure of how the distractor influences choice-related dynamics in the RNN.

To evaluate the robustness of this analysis window, we also computed the choice mode using the last 0.5 seconds and 1.5 seconds before the “go cue”. The results remained consistent, confirming that our findings are not dependent on the specific timing of the analysis window. Specifically, when using the last 0.5 seconds, the mean \pm SEM of the trial-switch fraction in control and APP-KI RNNs were $15.23 \pm 5.90\%$ and $88.30 \pm 5.61\%$, respectively ($P < 0.001$, $n = 30$ RNNs per group, one-tailed bootstrap). When using the last 1.5 seconds, the corresponding values were $49.50 \pm 8.59\%$ for control and $88.30 \pm 5.61\%$ for APP-KI RNNs ($P < 0.001$, $n = 30$ RNNs per group, one-tailed bootstrap). These consistent results demonstrate the robustness of our findings and support the conclusion that population activity along the choice axis was significantly less stable in APP-KI networks, regardless of the analysis window used. These results are now presented in **Figure S5c**.

We also re-analyzed the effects of reduced functional connectivity on network stability in control RNNs. When using the last 0.5 seconds to define the choice mode, the mean \pm SEM trial-switch fractions were $15.23 \pm 5.90\%$, $48.90 \pm 7.53\%$, and $76.62 \pm 5.97\%$ for 0%, 10%, and 20% ablation, respectively ($P < 0.001$, $n = 30$ RNNs per condition, one-way repeated-measures ANOVA). When using the last 1.5 seconds, the corresponding values were $49.50 \pm 8.59\%$, $81.56 \pm 5.92\%$, and $92.51 \pm 3.43\%$, respectively ($P < 0.001$, $n = 30$ RNNs per condition, one-way repeated-measures ANOVA). These consistent patterns across time windows further support the robustness of our findings and reinforce the conclusion that reduced functional connectivity contributes to unstable network dynamics underlying short-term memory maintenance. These results are also included in **Figure S5c**.

17. Line 199, how was 30 RNNs selected? A robustness analysis is needed.

We trained 30 RNNs for the control group and 30 RNNs for the APP-KI group, with each model initialized using a different random seed. We have included all trained models in the final analysis.

To test the robustness of our findings, we also re-analyzed the data using a smaller subset of models (11 RNNs per group), which more closely reflects the sample size in our actual experiments. Specifically, the mean \pm SEM of the trial-switch fraction in control and APP-KI RNNs were $25.82 \pm 11.66\%$ and $95.27 \pm 3.57\%$, respectively ($P < 0.001$, $n = 11$ RNNs, one-tailed bootstrap). In the control RNNs, the mean \pm SEM of the trial-switch fraction were $25.82 \pm 11.66\%$, $64.07 \pm 12.04\%$, and $75.56 \pm 12.57\%$ for 0%, 10%, and 20% ablation, respectively ($P < 0.001$, $n = 11$ RNNs, one-way repeated-measures ANOVA). These results were consistent with those obtained using all 30 RNNs, further supporting the robustness and reliability of our conclusions. The results are now presented in **Figure S5d**.

18. Paragraph starting on Line 202. Why was 1 RNN selected? A more direct approach would be to use multiple RNNs to mimic the multi-region dataset from their 2P work. Then, for each cortical region, this RNN analysis would be compared to the experimental data (for instance, is ALM/M2 or PPC the major contributors?). Also, the work about reducing functional connectivity is more applicable/insightful in a multi-region RNN model.

We thank the reviewer for these insightful comments. In this analysis, we focused on one type of RNN (control RNNs) to isolate the mechanistic impact of reduced functional connectivity on network dynamics. We observed that APP-KI RNNs were less stable in response to sensory perturbations compared to control RNNs (**Figures 5d and e**). We hypothesized that this instability could arise from reduced functional connectivity in APP-KI models, consistent with our observations in APP-KI mice (**Figures 4a and S4b**). To test this, we systematically ablated recurrent connections in control RNNs and found that increasing the proportion of ablated connections (from 10% to 20%) led to a higher probability of choice switching (**Figures 5f, g, S5c and d**).

We would like to clarify that our current approach already incorporates multi-regional data. Each RNN consists of 1024 units, trained to mimic the activity of 1024 neurons, with 128 neurons randomly sampled from each of the 8 cortical regions. Compared to training multiple region-specific RNNs, this method preserves both global network dynamics and local region-specific interactions.

Constructing RNN models with discrete modules, by contrast, would require assumptions about how information flows across the eight regions. While we agree that this is an important direction for future research, we believe it is beyond the scope of the current study.

19. Line 218, the inter-regional interactions require deeper analyses. Focus on the ones particularly relevant for APP observations and describe the differences in these results between control vs. APP.

Although not directly related to the main conclusion, we performed a statistical analysis of the optimal dimensionality within source regions that best predicts the activity of target regions. In APP-KI mice, we found that higher-dimensional representations of source regions were generally required to explain downstream activity (**Figure S6b**). These results are consistent with the interpretation that functional connectivity is globally altered in APP-KI mice.

20. Line 233, the hypothesis is extremely weak/brief. This is a critical transition point between experiments in their study and should be convincing to the reader. Improve and similarly across the Results section, double check to improve other hypotheses/transitions.

We apologize for the lack of clarity. We have revised the text as follows: “The heightened vulnerability of APP-KI mice to distractors suggests reduced robustness in their neural network dynamics. One potential contributor

to this vulnerability is diminished degeneracy in inter-regional communication (Fornito., et al., 2015). Degeneracy describes the capacity of distinct elements within a system to perform the same function, thereby providing functional robustness (Tononi et al., 1999). In particular, spatial degeneracy refers to the presence of multiple parallel pathways capable of performing the same function, providing flexibility and resilience when one pathway is compromised.

To determine whether APP-KI mice show reduced spatial degeneracy, we examined how similar neural activity patterns within a source region's communication subspace are routed to different downstream target regions (**Figure 6a**). High spatial degeneracy arises when similar activity fluctuations are transmitted to multiple targets, ensuring robust sensorimotor transformations even when one pathway is perturbed (**Figure 6b**). We hypothesized that reduced spatial degeneracy in APP-KI mice, leading to compromised compensatory mechanisms, underlies their short-term memory vulnerability.”. This revision is now included in lines 230–241 on page 7.

21. Lines 239-241, in addition to listing regions that were significantly different, some insightful deeper observations are needed. Without observations, the impact of these experiments/results are poor.

We thank the reviewer for raising this point. In response, and building on our previous work demonstrating an important role of PPC in enhancing the robustness of ALM neural activity in a non-disease mouse model (Chia *et al.*, 2023), we focused on PPC as a source region and computed subspace similarities between ALM and PPC. We found that subspace similarity was higher in control mice than in APP-KI mice during the delay epoch (lines 248-252 on page 8). This additional analysis reinforces the idea that PPC conveys similar information to both itself and ALM during short-term memory.

22. Fig 6E, typo in the panel?

We carefully examined the original **Figure 6e** and did not find a typographical error in the panel. If the reviewer could clarify the specific issue they are referring to, we would be happy to make the necessary corrections.

23. Line 247 is too brief. Expand and thoroughly discuss the results, not just a brief/general statement.

We thank the reviewer for this valuable feedback. In response, we have expanded the text (now lines 257–260 on page 8 of the revised manuscript) to provide a more detailed explanation of the findings: “In both control and APP-KI mice, removing source activity along dimensions that best predicted a given target reduced prediction performance for the same target (**Figures 6g and h**). Importantly, prediction performance also declined when

the predicted region differed from the target, suggesting that predictive dimensions in the source region were shared across multiple downstream regions (**Figures 6g and h**).”.

24. For Fig S5C (and Fig 6F), the main text does not discuss the 1 vs. 3 predictive dimension removal results at all. This is unacceptable.

We have now included the results related to the removal of 1 and 3 predictive dimensions in lines 264 (page 8) of the revised manuscript.

25. Fig 6G should also be provided for the removal of 1 or 3 predictive dimensions, to contrast with the current figure panel.

These results were already presented in the previous version of the manuscript. **Figure S6d** complements the analysis in **Figure 6g** by enabling direct comparison of prediction performance across different levels of dimensional removal.

26. Line 263, is there some literature that led to this hypothesis? How did the authors think about this, what papers influenced them? Those should be described and cited, as supporting evidence for this idea.

We thank the reviewer for this important feedback. Previous studies have shown that although population activity patterns evolve dynamically during short-term memory, downstream regions can still extract stable stimulus information from a low-dimensional subspace (Murray et al., 2017; Parthasarathy et al., 2019). This phenomenon, often referred to as temporal degeneracy, provides redundancy that helps maintain robust information flow despite distractors and noise. We therefore reasoned that the observed vulnerability of short-term memory in APP-KI mice reflects a reduction in temporal degeneracy: as the system loses this compensatory capacity, it becomes less resilient to perturbations (Barulli and Stern, 2013). We have added this explanation in lines 270-273 (page 8).

27. Line 267, 1 sec interval needs a justification and robustness analysis.

We thank the reviewer for this suggestion. The 1-second interval was chosen based on the duration of the stimulus epoch, which is 1 second and represents the shortest among all task epochs. Using this window allows us to capture transitions in subspace similarity across task epochs (e.g. from ITI to stimulus to delay). A longer window (e.g. 2 seconds) would reduce temporal resolution, potentially obscuring rapid transitions in subspace similarity between adjacent task epochs and making them less interpretable. Conversely, a shorter window (e.g.

0.5 seconds) may yield redundant results, as subspace similarity generally remains stable within each task epoch (as supported by our delay-epoch results in **Figure 7d**). We therefore believe the 1-second interval provides an optimal balance between temporal resolution and interpretability.

Given this conceptual alignment between the 1-second window and task structure, as well as the expected limitations of shorter and longer intervals, we did not conduct an exhaustive robustness analysis across multiple window sizes. The rationale outlined above reflects our consideration of the trade-offs involved in this choice.

28. Lines 269-273, quantify for control vs. APP on a per region basis, and plot the results clearly to support these observations.

The subspace similarity across time for control and APP-KI mice on a per-region basis is shown in **Figures 7d, e and S7a-d**. The figures displays subspace similarity values evaluated across seven time points for both groups. At each time point, similarity is visualized as a 2D heatmap matrix, where each entry reflects the similarity between a pair of cortical regions. Columns (left to right) and rows (top to bottom) correspond to the following regions: ALM, M1a, M1p, M2, S1fl, vS1, RSC and PPC, as schematized in **Figure 7c**. **Figure 7e** further summarizes these results on a per-region basis, showing intra-regional subspace similarity in the left panel and inter-regional subspace similarity in the right panel for both control and APP-KI mice, with different colors denoting different cortical regions.

The findings described in lines 284–293 on page 8–9 of the revised manuscript can be directly identified from these regional results (**Figures 7d, e and S7a-d**). First, subspace similarity was higher when the source and target regions were the same (diagonal) compared with across (non-diagonal) regions. Second, subspace similarity was greatest between adjacent trial epochs. Third, posterior cortical regions exhibited higher subspace similarity, particularly in control mice. Fourth, across all conditions, subspace similarity was consistently lower in APP-KI than in control mice.

29. Fig 7D, grey scale shades are very hard to distinguish. Either use different colors or separate graphs in addition to these to improve clarity for the reader.

We agree with the reviewer that the grayscale shades in **Figure 7d** (now **Figure 7e**) were difficult to distinguish. To improve clarity, we have revised the figure by using distinct colors for each condition.

30. Fig 7E, another example of a poorly justified and described result. Improve text. Add deeper implications of gradual vs. abrupt for STM function.

We thank the reviewer for this valuable feedback. In response, we revised the Results section to clarify group differences in **Figure 7e** (now **Figure 7f**) and elaborate on their implications for short-term memory. Specifically, we added the following text at lines 293–297 (page 9) of the revised manuscript: “In addition, during the stimulus and delay periods, control mice exhibited gradual, continuous transitions in activity patterns, whereas APP-KI mice showed more abrupt shifts (**Figure 7f**). Together, these results suggest that APP-KI mice have reduced temporal degeneracy, with fewer redundant or overlapping activity trajectories supporting the same memory across time. Such abrupt transitions in APP-KI mice likely contribute to their heightened susceptibility to perturbations and short-term memory deficits.”

31. The Discussion section is unacceptable. There is no deeper contextualization of their results while taking into account basic science findings in the field for the same regions studied in this paper, as well as from the AD literature. Significant effort and improvement is needed.

We agree that the Discussion section in the previous version was insufficient. In response, we have thoroughly revised and expanded the entire Discussion section.

32. Fig S1B, bottom graphs need additional labeling in the figure itself and the legend, to clearly indicate what each plot refers to. The reader shouldn't need to be guessing or deciding themselves.

We have revised the original **Figure S1b** (now **Figure S2b**) by including additional labels in the figure panels and updating the figure legend.

Reference

Chia, X.W., Tan, J.K., Ang, L.F., Kamigaki, T., and Makino, H. (2023). Emergence of cortical network motifs for short-term memory during learning. *Nat Commun* 14, 6869. 10.1038/s41467-023-42609-4.

Saito, T., Matsuba, Y., Mihira, N., Takano, J., Nilsson, P., Itohara, S., Iwata, N., and Saido, T.C. (2014). Single App knock-in mouse models of Alzheimer's disease. *Nature Neuroscience* 17, 661-663. 10.1038/nn.3697.

Murray, J.D., Bernacchia, A., Roy, N.A., Constantinidis, C., Romo, R., and Wang, X.J. (2017). Stable population coding for working memory coexists with heterogeneous neural dynamics in prefrontal cortex. *Proc Natl Acad Sci U S A* 114, 394-399. 10.1073/pnas.1619449114.

Parthasarathy, A., Tang, C., Herikstad, R., Cheong, L.F., Yen, S.C., and Libedinsky, C. (2019). Time-invariant working memory representations in the presence of code-morphing in the lateral prefrontal cortex. *Nat Commun* 10, 4995. [10.1038/s41467-019-12841-y](https://doi.org/10.1038/s41467-019-12841-y).

Tononi, G., Sporns, O., and Edelman, G.M. (1999). Measures of degeneracy and redundancy in biological networks. *Proc Natl Acad Sci U S A* 96, 3257-3262. [10.1073/pnas.96.6.3257](https://doi.org/10.1073/pnas.96.6.3257).

Reviewer #3 (Remarks to the Author):

In this manuscript, “Vulnerability of short-term memory in a mouse model of Alzheimer’s disease”, Li et al. seek to determine the aberrant neural mechanisms underlying short-term memory deficits observed in Alzheimer’s disease (AD). They hypothesize that this deficits in short-term memory are caused by the decreased stability of their recurrent neural network activity. Recurrent neural network is thought to support the sustained neural activity in response to a brief stimulus which is considered to be the neural correlates of short-term memory. Employing two-photon microscope calcium imaging in a mouse model of AD (APP-KI) during a delayed-response tactile discrimination task, the authors report that the neural selectivity in these mice is more susceptible to sensory distractors at both single-neuron and population levels. They observed reduced functional connectivity across dorsal cortex as well as weak spatiotemporal degeneracy, which they concluded to be the central mechanisms underlying the short-term deficits in these mice. The study addresses an important mechanistic question regarding one of the most prevalent neurodegenerative diseases, and the manuscript is written nicely with clear organization. However, I have several concerns that I would like to see addressed. My major and minor concerns are described below.

We thank the reviewer for their valuable feedback on our manuscript. In the revised manuscript, we have incorporated previous findings to illustrate the correlation between APP-KI mouse age and A β plaque accumulation. We have also updated the figures to present general neuronal activity in APP-KI mice and added detailed information describing the data analysis methods used to match neural activity between control and APP-KI mice. We believe these revisions have substantially strengthened the manuscript. For clarity, all major changes have been highlighted in yellow in the revised manuscript.

Major concerns:

1. The age of the mice used in the study is quite variable ranging from 6 to 10 months. This is of some concern when the study involves a model for a progressive neurodegenerative disease. What is the age for the expected onset of AD phenotype in APP-KI mice? How does it relate to the range of subject age used in the study? Does the A β plaque accumulation similar across these 5 months? If not, did the severity of the reported observations correlate with age or the amount of A β plaques observed in APP-KI mice?

We thank the reviewer for raising this important point. In APP-KI mice, A β accumulation begins around 2 months of age and reaches near-saturation levels by approximately 7 months (Saito et al., 2014). Behavioral deficits in these mice have also been reported to occur around this age. In our study, the ages of APP-KI mice ranged from 6 months and 25 days to 9 months and 28 days. This range was selected specifically to ensure that all animals

had reached a stage of robust and relatively stable A β plaque burden. Therefore, we expect no significant variability in plaque deposition across the APP-KI mice used in our experiments. We have added this explanation in lines 79–81 (page 3) of the revised manuscript.

2. Throughout the manuscript, there is very little opportunity to see the raw data. Figure 1D shows some raster plots of task-related activity from sample cells, but it appears that those are only from the wildtype mice.

The example neurons shown in **Figure 1d** are from an APP-KI mouse. We apologize for not making this clear in the original version and have now added this information to the figure legend.

What do the general neuronal activity look like in APP-KI mice? What are the total number of active neurons for each group that were used for the analysis shown in Figure 1E?

Example neural activity in APP-KI mice is shown in **Figure 1d**. In control mice, the total number of neurons used for **Figure 1e** were: ALM: 1395; M1a: 2186; M1p: 7593; M2: 1876; S1fl: 3916; vS1: 1847; RSC: 3582; PPC: 3886 neurons from 32 sessions, 7 mice. In APP-KI mice, the corresponding totals were: ALM: 2586; M1a: 4988; M1p: 9478; M2: 4018; S1fl: 6922; vS1: 3554; RSC: 6650; PPC: 7854 neurons from 40 sessions, 11 mice. These values are reported in lines 709–713 (page 38) of the revised manuscript.

I think the big elephant in the room that is not addressed in the manuscript is the possibility that neural activity in APP-KI is reduced in general. If that is the case, then it will contribute as a confound to the subsequent network analysis like connectivity, correlation, and regression. The criterion for including the neurons in analysis provided in the method section is very broad: “Neurons whose z-scored deconvolved calcium signal exceeded 10 at least once every 10 min were included for analysis”. In the text, it is stated that “neurons were subsampled to match activity levels between control and APP-KI mice to avoid them serving as a potential confounding variable”, but no details as to how this was achieved is provided.

Figure 2C shows some activity in control and APP-KI mice, but the activity levels are normalized to each cells own maximum and minimum levels, making it difficult to compare among neurons or across groups. Given the limited number of cells that used as criterion for including each cortical region in the analyses (regions were included if they contained at least 20 neurons for intra-regional and 10 neurons for inter-regional analysis, respectively), the lack of clear assessment of the general activity levels of APP-KI is concerning.

We agree that neural activity level is an important factor that could influence downstream analyses such as connectivity, correlation, and regression, and we thank the reviewer for highlighting this point. We first applied

a broad inclusion threshold (z-score > 10 at least once every 10 min) to exclude completely inactive neurons while retaining those with sparse but potentially task-relevant activity. To more rigorously control for group-level differences in baseline activity, we then applied a second, more selective filtering step using preset activity thresholds. Specifically, for each brain region and session, we computed the mean activity level of each neuron by averaging its z-scored activity across relevant time windows. For functional connectivity, this was calculated across all concatenated inter-trial interval (ITI) epochs within a session. For correlation and regression analyses, we used time frames from -1 to 7 s relative to trial onset. Threshold values were empirically determined to ensure that the resulting distributions of mean activity across neurons were closely matched between groups (see overlap coefficients below). These procedures are now described in lines 819–822, 829–832, 835–838 (page 41) of the revised manuscript.

We also agree that providing a clear assessment of general activity levels across groups is important. In response, we included histograms showing the distribution of mean activity levels across neurons for each group. The overlap coefficients between control and APP-KI mice were 0.94 (ITI) and 0.89 (trial frames), where 1 indicates identical distributions. These values indicate that most neurons in both groups exhibited comparable baseline activity levels. The results are now included in **Figure S4a**.

3. Also relating to my concern #2, the hypothesized secession of persistent neural activity as the cause of short-term memory impairment is not clearly demonstrated since the relevant data regarding perturbations by distractors provided in the manuscript have been process to “decoding accuracy”. What does data presented in Figure 2C look like during the trials that had early, middle, and late distractors?

We thank the reviewer for the question and apologize for any confusion. The data corresponding to trials with early, middle and late distractors were already included in the original submission (now in **Figures 2e and S2a**). For reference, **Figure 2c** shows normalized trial-averaged activity of left- and right-selective delay neurons during correct no-distractor trials in control and APP-KI mice, illustrating the trial-type specificity of these neurons.

To show how this activity is modulated under distractor conditions, we visualized the same neural population’s dynamics during distractor trials using a more concise representation. Specifically, we computed the trial-type selectivity profile for each delay neuron by subtracting its activity in left trials (with or without distractors) from the corresponding right trials (with or without distractors). This method yields upward and downward deflections for right- and left-selective delay neurons, respectively, in a format that allows clearer comparison across distractor timings. We then averaged these trial-type selectivity profiles across neurons within each brain region and plotted the mean \pm SEM over time. These results are presented in **Figures 2e and S2a**, providing a direct view of how early, middle and late distractors alter trial-type selectivity. The corresponding analysis methods are described in lines 724–739 (page 38–39) of the revised manuscript.

Figure. Distractor-mediated neural activity modulations

Z-scored activity of left- and right-selective delay neurons during correct distractor trials in control and APP-KI mice, along with baseline-subtracted mean traces (baseline defined as the first 0.5 s of the trace). Neurons within each population were sorted by their peak activity timing. Notably, APP-KI mice exhibited smaller separations between mean left- and right-trial activities following distractors compared with control mice.

In addition, we performed a further analysis to address the reviewer’s last point (**Figure**). Unlike in **Figure 2c**, here we did not normalize the activity of each neuron, allowing us to capture distractor-driven modulations in raw activity. Our imaging regions were located in the left cortex, and we observed that right-trial-selective delay neurons in APP-KI mice exhibited greater vulnerability to distractors, as reflected by the decreased separation between their mean activity in left and right trials following distractor presentation.

4. Figure 3B, D, G, I: Are these results from the pseudo-mice generated with random sampling of subsets of neurons from all 8 cortical regions combined or are they from a specific cortical region? If the former, I am surprised that they seem to capture the perturbation only present in a few (two: ALM and M2) of the eight cortical regions? I would appreciate a little more in-depth explanation of these findings.

We thank the reviewer for these comments and apologize for any confusion. For **Figures 3b and g** (now **Figures 3b and h**), the results are based on pseudo-mice generated by random sampling of subsets of neurons pooled from all 8 cortical regions. These analyses revealed that, compared with control mice, APP-KI mice exhibited significantly greater sensitivity to early, middle and late distractors, as quantified in **Figures 3c and 3h** (now **Figures 3c and 3i**). The corresponding sampling methods are described in lines 745–747 (page 39), lines 759–760 (page 39) and lines 789–790 (page 40) of the revised manuscript.

For **Figures 3d and 3i** (now **Figure 3d**), the results were also based on pseudo-mice sampled from all 8 cortical regions. However, to assess regional contributions, we performed an ablation analysis; for each condition, neural activity from individual cells in the region of interest was retained, while neural activity in all other regions was replaced with their mean activity. This approach allowed us to isolate the functional relevance of each region for distractor-related decoding performance. The corresponding methods are described in lines 805–811 (page 40) of the revised manuscript.

Previous **Figures 3d and 3i** (now **Figure 3d**) depict the temporal profile of distractor-induced decoding accuracy relative to the no-distractor condition. The results were quantified in **Figure 3e**, where lower values reflect increased regional vulnerability to distractors, whereas higher values reflect preserved robustness. In both control and APP-KI mice, multiple regions exhibited reduced decoding accuracy in the presence of distractors, indicating that distractor sensitivity is distributed across cortical areas rather than confined to ALM and M2. Control mice showed significantly greater robustness than APP-KI mice in ALM and M2. In other regions, decoding accuracy was generally higher in the control mice as well, although these differences did not reach statistical significance (one-tailed bootstrap with an FDR with the Benjamini-Hochberg procedure). The stronger contribution of ALM and M2 reflects their carrying highly task-relevant signals and being particularly perturbed by distractors, consistent with **Figures 2e, f, S2a and b**.

5. Figure 4: It seems to this reviewer that all three measurements, functional connectivity, trial-by-trial correlation, regression would be critically impacted by overall activity levels, but that seems to be the one metric that is not clearly described in this manuscript.

Please refer to our detailed response to Comment #2. Briefly, we included histograms showing the distribution of mean neural activity across neurons in control and APP-KI mice. These distributions substantially overlapped, with overlap coefficients of 0.94 for ITI-based activity and 0.89 for trial-based activity (where 1 indicates identical distributions). These results suggest that overall activity levels were comparable between groups and are now presented in **Figure S4a**.

6. I was intrigued by the data shown in Figure 5G where the fraction of the trials that switched decisions in 20% ablation of connectivity in the controls was comparable to those in APP-KI. Does this 20% ablation resemble the decreased % of neurons exhibiting epoch-specific selectivity in APP-KI (Figure 1E)?

We thank the reviewer for this insightful comment. With the 20% synaptic ablation, our aim was not to precisely replicate the reduced proportion of neurons exhibiting task-epoch-specific selectivity shown in **Figure 1e**. While the similar level of reduction in trial-selective neurons may indeed contribute to impaired memory-related representations, establishing a direct mechanistic correspondence between the two will require further investigation in future studies.

7. Figure 7, Is the temporal degeneracy the population code correlate of “persistent activity” that the authors hypothesized to support short-term memory?

Yes, by temporal degeneracy we refer to the redundant representation of information by neural populations across time, such that the dynamic evolution of population activity trajectories implements what was classically described as “persistent activity”. In APP-KI mice, synaptic deficits likely reduce this redundancy and, as a result, the robustness of population-level representations of stimulus and choice. We have added this clarification in lines 270–273 (page 8).

Minor concerns

1. The immunostaining of A β plaques shown in Figure 1C is too small to see and should be accompanied by the staining in control mice of matched age.

We have updated **Figure 1c** by enlarging the image of A β plaque staining to improve visibility. In addition, we now include staining from an age-matched control mouse in **Figure S1c** to enable direct comparison of A β plaque deposition.

2. Figure S2, decoding accuracy looks better for APP-KI mice compared to the controls in the absence of distractors. Is that the case?

Decoding accuracy in the absence of distractors appears slightly higher in APP-KI mice compared with control mice (**Figures S3a and g**). However, because our analyses focus on relative decoding accuracy, which controls for differences in the no-distractor condition, this subtle baseline difference does not affect our conclusions. Using this measure, we found that APP-KI mice exhibited markedly greater impairments in decoding accuracy when distractors were present (**Figures 3b, c, h and i**).

Reference

Saito, T., Matsuba, Y., Mihira, N., Takano, J., Nilsson, P., Itohara, S., Iwata, N., and Saido, T.C. (2014). Single App knock-in mouse models of Alzheimer's disease. *Nature Neuroscience* 17, 661-663. 10.1038/nn.3697.

Reviewer #1 (Remarks to the Author):

The authors have satisfactorily addressed all of my previous comments and concerns.

We thank the reviewer for their thoughtful evaluation and appreciate their recommendation for publication.

Reviewer #2 (Remarks to the Author):

The authors did a great job addressing my comments from round 1. Please see below for a few final thoughts:

We appreciate the reviewer's constructive feedback.

Previous point 4. Please quantify Thy1-GCaMP-positive cell numbers at the specific ages of mice used and for each cortical region investigated in this study (with statistics). --- While they provided this information in their rebuttal, please confirm that it is included in the revised manuscript somewhere.

The results are now included in the main text and Methods section on lines 91–96 (page 3) and lines 725-735 (page 39) of the revised manuscript.

Previous point 5. Line 94, the delay period of 4 sec is quite long to simply average out in their analyses. What would be found if the authors separated early vs. late delay periods? --- Their response sounds good, but they need to discuss in the main text what it could mean that vS1 is significant in the initial delay period, along with what appears to be a strong trend in the later delay period too (Fig S1e). Could this difference add to those contributed by changes in PPC?

We thank the reviewer for this comment. In addition to the significant reduction in vS1 selectivity during the initial delay period and a strong trend toward reduced selectivity later in the delay, we also observed a strong tendency for reduced vS1 selectivity during the stimulus period (**Figures 1e and S1e**). These results suggest that selectivity was already degraded at the input stage and further amplified in the PPC of APP-KI mice. The corresponding descriptions are provided on lines 96–102 (pages 3-4) of the revised manuscript.

Previous point 22. Fig 6E, typo in the panel? --- The typo is in the word "Predication".

We thank the reviewer for catching this typo. This has been corrected in the revised manuscript.

Reviewer #3 (Remarks to the Author):

This is a revised manuscript entitled “Vulnerability of short-term memory in a mouse model of Alzheimer’s disease” in which Li et al., examine the aberrant recurrent neural network mechanism underlying short-term memory deficits observed in AD. In response to the previous round of reviews, the authors addressed most if not all of my comments adequately. I have but one lingering comment on the overall activity levels of the mutant vs control APP mice that were used in the study. While I appreciate the authors’ effort to provide the mean activity level distribution histogram for each group, general overlap coefficient is not the appropriate measure in this case. I say this because the histograms provided for “trial” (Figure S4a bottom) seems to reflect my original concern, which is that activity levels of APP mutants may be lower compared to that of the controls. I recommend that authors run a KS Test on these two distributions to address this remaining concern.

We thank the reviewer for their constructive feedback. Although we had previously tested whether such differences existed, we identified an error in the previous version of our response related to this point, and this has now been corrected in the revised manuscript. We sincerely apologize for the confusion. This correction does not affect the results relevant to this point or the conclusions of the study. As before, using the KS test, we confirmed that there are no statistical differences between control and APP-KI mice in the probability distributions of mean z-scored neural activity during the trial and ITI, both across all regions combined and within individual regions (**Figure S4a**, lines 569–576 on pages 28-29).